# Pan-genome analysis identifies intersecting roles for *Pseudomonas* specialized metabolites in potato pathogen inhibition

Alba Pacheco-Moreno[1†], Francesca L Stefanato[1†], Jonathan J Ford[1], Christine Trippel[1], Simon Uszkoreit[1], Laura Ferrafiat[1], Lucia Grenga[1], Ruth Dickens[1], Nathan Kelly[1], Alexander DH Kingdon[1], Liana Ambrosetti[1], Sergey A Nepogodiev[2], Kim C Findlay[3], Jitender Cheema[4], Martin Trick[4], Govind Chandra[1], Graham Tomalin[5], Jacob G Malone[1,6]*, Andrew W Truman[1]*

[1]Department of Molecular Microbiology, John Innes Centre, Norwich, United Kingdom; [2]Department of Biochemistry and Metabolism, John Innes Centre, Norwich, United Kingdom; [3]Department of Cell and Developmental Biology, John Innes Centre, Norwich, United Kingdom; [4]Department of Computational and Systems Biology, John Innes Centre, Norwich, United Kingdom; [5]VCS Potatoes, 2 Burnt Cottages, Framlingham, United Kingdom; [6]School of Biological Sciences, University of East Anglia, Norwich, United Kingdom

*For correspondence:
jacob.malone@jic.ac.uk (JGM);
andrew.truman@jic.ac.uk (AWT)

†These authors contributed equally to this work

**Abstract** Agricultural soil harbors a diverse microbiome that can form beneficial relationships with plants, including the inhibition of plant pathogens. *Pseudomonas* spp. are one of the most abundant bacterial genera in the soil and rhizosphere and play important roles in promoting plant health. However, the genetic determinants of this beneficial activity are only partially understood. Here, we genetically and phenotypically characterize the *Pseudomonas fluorescens* population in a commercial potato field, where we identify strong correlations between specialized metabolite biosynthesis and antagonism of the potato pathogens *Streptomyces scabies* and *Phytophthora infestans*. Genetic and chemical analyses identified hydrogen cyanide and cyclic lipopeptides as key specialized metabolites associated with *S. scabies* inhibition, which was supported by in planta biocontrol experiments. We show that a single potato field contains a hugely diverse and dynamic population of *Pseudomonas* bacteria, whose capacity to produce specialized metabolites is shaped both by plant colonization and defined environmental inputs.

## Editor's evaluation

This work uses large-scale genome sequencing and analysis, mass spectrometry, and bioassays to investigate the genomic diversity of *Pseudomonas* strains and their role in plant protection. The authors identified key metabolites that inhibit *Streptomyces scabies*, the causal agent of potato scab, and showed how genomic diversity in closely related bacterial strains can contribute to plant pathogen suppression in the field.

## Introduction

Plant pathogenic microorganisms are responsible for major crop losses worldwide and represent a substantial threat to food security. Potato scab is one of the main diseases affecting potato quality

**eLife digest** Potato scab and blight are two major diseases which can cause heavy crop losses. They are caused, respectively, by the bacterium *Streptomyces scabies* and an oomycete (a fungus-like organism) known as *Phytophthora infestans*.

Fighting these disease-causing microorganisms can involve crop management techniques – for example, ensuring that a field is well irrigated helps to keep *S. scabies* at bay. Harnessing biological control agents can also offer ways to control disease while respecting the environment. Biocontrol bacteria, such as *Pseudomonas*, can produce compounds that keep *S. scabies* and *P. infestans* in check. However, the identity of these molecules and how irrigation can influence *Pseudomonas* population remains unknown.

To examine these questions, Pacheco-Moreno et al. sampled and isolated hundreds of *Pseudomonas* strains from a commercial potato field, closely examining the genomes of 69 of these. Comparing the genetic information of strains based on whether they could control the growth of *S. scabies* revealed that compounds known as cyclic lipopeptides are key to controlling the growth of *S. scabies* and *P. infestans*. Whether the field was irrigated also had a large impact on the strains forming the *Pseudomonas* population.

Working out how *Pseudomonas* bacteria block disease could speed up the search for biological control agents. The approach developed by Pacheco-Moreno et al. could help to predict which strains might be most effective based on their genetic features. Similar experiments could also work for other combinations of plants and diseases.

(*Larkin et al., 2011*) and presents a significant economic burden to farmers around the world. The Gram-positive bacterium *Streptomyces scabies*, which is the causal organism of potato scab, is ubiquitous and presents a threat in almost all soils (*Bignell et al., 2010*; *Lerat et al., 2009*). Properly managed irrigation is a reasonably effective control measure for potato scab. However, scab outbreaks still regularly occur in irrigated soil, and with increasing pressures on water use it is clear that alternative approaches to the control of scab are needed. An attractive potential alternative involves the exploitation of soil microorganisms that suppress or kill plant pathogens, known as biocontrol agents (*Köhl et al., 2019*; *Weller, 2007*).

Many soil-dwelling *Pseudomonas* species form beneficial relationships with plants, positively affecting nutrition and health (*Cheng et al., 2017*; *Loper et al., 2012*; *Zamioudis et al., 2013*) and exhibiting potent antagonistic behavior towards pathogenic microorganisms (*Biessy et al., 2019*; *Haas and Défago, 2005*; *Weller, 2007*). *Pseudomonas* influence the plant environment using a diverse range of secondary metabolites (*Arseneault et al., 2013*; *Gross and Loper, 2009*; *Nguyen et al., 2016*; *Stringlis et al., 2018*) and secreted proteins (*Ghequire and De Mot, 2014*; *Rangel et al., 2016*). As such, *Pseudomonas* sp. have been identified as key biocontrol organisms in numerous plant-microbe systems (*Mauchline et al., 2015*; *Wei et al., 2019*), and these bacteria have potential applications as agricultural biocontrol agents and biofertilizers (*Kwak and Weller, 2013*; *Weller, 2007*). Many soil pseudomonads belong to the *Pseudomonas fluorescens* group, which consists of over 50 subspecies and exhibits huge phenotypic and genetic diversity (*Biessy et al., 2019*; *Gomila et al., 2015*; *Loper et al., 2012*; *Melnyk et al., 2019*; *Silby et al., 2009*), with a core genome of about 1300 genes and a pan-genome of over 30,000 genes (*Garrido-Sanz et al., 2016*). These bacteria use a variety of mechanisms to colonize the plant rhizosphere (*Little et al., 2019*), communicate with plants (*Zamioudis et al., 2013*), and suppress a range of plant pathogens (*Haas and Défago, 2005*), including bacteria (*Arseneault et al., 2015*), fungi (*Michelsen et al., 2015*), and insects (*Flury et al., 2017*), although a single strain is unlikely to have all of these attributes. Specialized metabolites are critical to many of these ecological functions, and the *Pseudomonas* specialized metabolome is one of the richest and best characterized of any bacterial genus (*Gross and Loper, 2009*; *Nguyen et al., 2016*; *Stringlis et al., 2018*).

Various studies have associated pseudomonads with potato scab suppression. A significant increase in the abundance of *Pseudomonas* taxa has been observed for irrigated fields, correlating with reduced levels of potato scab (*Elphinstone et al., 2009*). Naturally scab-suppressive soils have also been shown to contain a greater proportion of *Pseudomonas* when compared to scab-conducive

soils (**Meng et al., 2012**; **Rosenzweig et al., 2012**), and phenazine production by *P. fluorescens* can contribute to scab control (**Arseneault et al., 2015**; **Arseneault et al., 2013**; **Arseneault et al., 2016**). Differences between soil microbial populations that enable effective pathogen suppression are routinely assessed using amplicon sequencing (**Fierer, 2017**; **Rosenzweig et al., 2012**). However, the heterogeneity of the *P. fluorescens* group limits the usefulness of these methods for observing changes at the species or even the genus level. To effectively determine the relationship between the soil *Pseudomonas* population and disease suppression, it is important to accurately survey genotypic and phenotypic variability at the level of individual isolates, and to determine how this variation is linked to agriculturally relevant environmental changes (**Mauchline and Malone, 2017**).

To investigate the genetic bases for *S. scabies* inhibition by *P. fluorescens* and to assess whether the scab-suppressive effects of irrigation derive from increased populations of biocontrol genotypes in the soil or on the plant, we focused on the *Pseudomonas* population from a potato field susceptible to potato scab. We first employed a phenotype-genotype correlation analysis across *P. fluorescens* strains isolated from a single potato field. We hypothesized that an unbiased correlation analysis would identify genetic loci and biosynthetic gene clusters (BGCs) that may be overlooked by screening for bioactive small molecules or by focusing on the biosynthetic repertoire of a limited number of strains. Here, we correlated phylogeny, phenotypes, specialized metabolism, and accessory genome loci, then investigated the importance of strong correlations by genetic manipulation of selected wild isolates. In total, 432 *Pseudomonas* strains were phenotyped (with 69 whole genomes sequenced). This approach also enabled us to answer a number of ancillary questions: how diverse is the *P. fluorescens* population from a single field location? Do the phenotypes associated with a *P. fluorescens* strain correlate with its biosynthetic capacity? What does irrigation do to both the population structure of the *P. fluorescens* group and to the wider bacterial community? Using this approach, we identify the *P. fluorescens* genes, gene clusters, and natural products that are required for potato pathogen suppression in vitro. We use this data to inform the discovery that the cyclic lipopeptide (CLP) tensin is a key determinant of in planta pathogen suppression by a *Pseudomonas* species. We show that irrigation induces profound and repeatable changes in the microbiome, both on a global level and within the *P. fluorescens* species group. Finally, we propose a model for the relationship between irrigation, pathogen suppression, and population-level shifts within the plant-associated *P. fluorescens* population.

## Results

### Irrigation induces a significant change in the soil microbiome

The ability of irrigation to protect root vegetables against *S. scabies* infection is agriculturally important and widespread, but poorly understood. It is likely that the irrigated soil microbiota plays a role in mediating scab suppression, but how this occurs is unclear. We therefore assessed the impact of irrigation on the total bacterial population of a commercial potato field in the United Kingdom. Multiple soil samples were taken from two sites (A1 and B1) within this field, immediately prior to potato planting in January. Following potato planting, one site was irrigated as normal (site A), while the second was protected from irrigation (site B). Tuber-associated soil was sampled from both sites in May (A2 and B2) when tubers were just forming, and the plants were most susceptible to *S. scabies* infection. Total genomic DNA was then extracted from replicate samples of each site after each sampling event, and 16S rRNA amplicon sequencing was used to examine the bacterial population in each of these sites (*Figure 1*).

We used voom (**Law et al., 2014**) with a false discovery rate (FDR) of 0.05 to assess population changes across the four sampling sites. In total, changes were observed for 26 bacterial orders (*Figure 1A*), with the most significant changes observed between January and May regardless of irrigation. This partially reflects an increase in bacterial orders that have previously been associated with the potato root microbiome (**Pfeiffer et al., 2017**; **Weinert et al., 2011**), including *Rhizobiales*, *Sphingomonadales,* and *Flavobacteriales*. Significant population changes (FDR < 0.05) were also observed for eight bacterial orders between the irrigated (A2) and nonirrigated (B2) sites (*Figure 1B and C*), including a larger proportion of *Pseudomonadales* bacteria in the irrigated site. In contrast, despite the potential for microbial heterogeneity across the fertilized field prior to planting, no significant changes were observed between pre-planting sites A1 and B1.

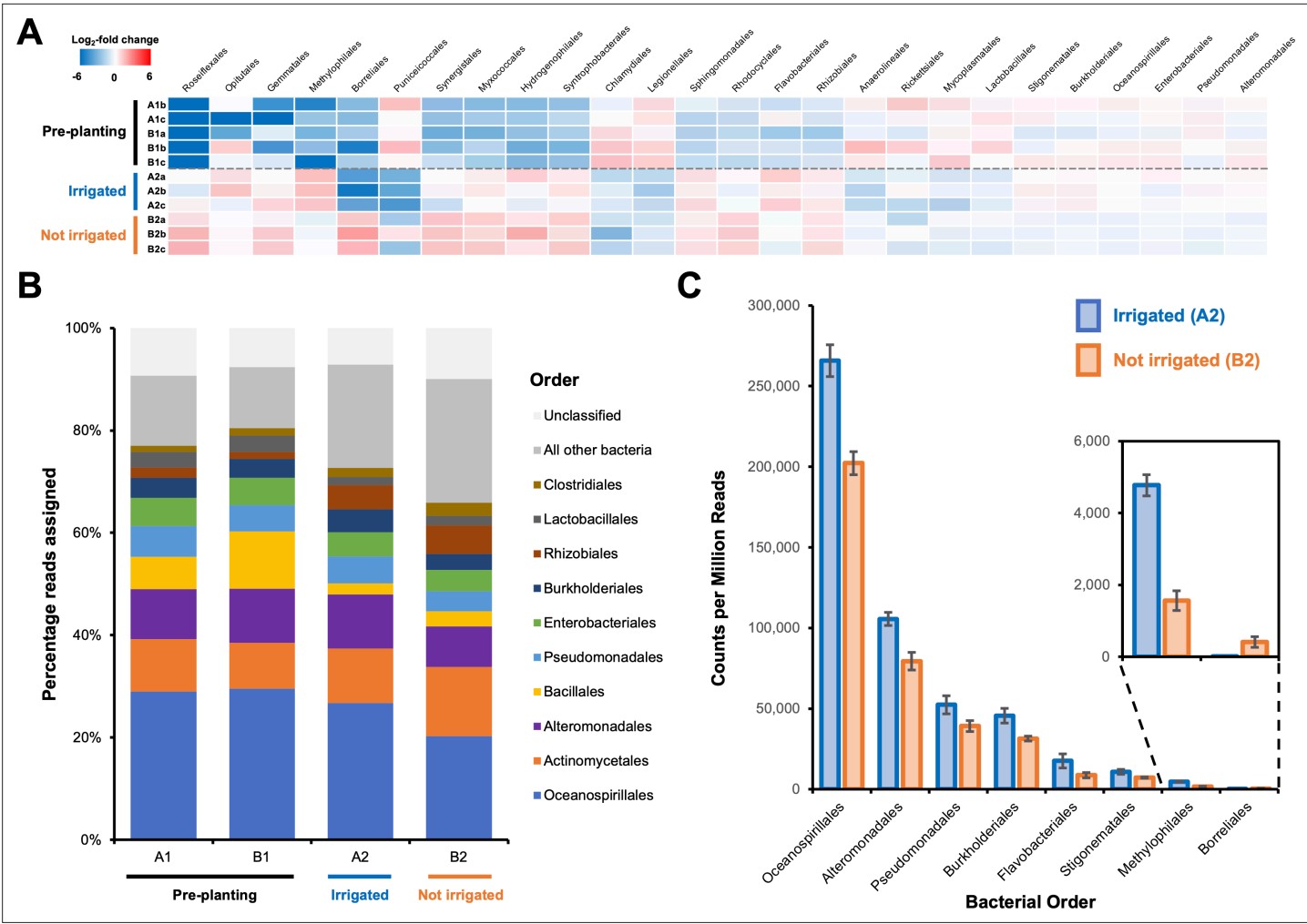

**Figure 1.** Effect of irrigation on the microbial population of a potato field. (**A**) The 26 bacterial orders whose populations were determined to significantly differ across one or more sampling sites using voom with a false discovery rate of 0.05. Data are shown as a heatmap of the $\log_2$-fold change with respect to the overall average counts per million for a given order. Sample A1a was omitted from the analysis due to possible contamination leading to an atypical bacterial population (*Figure 1—figure supplement 1*). (**B**) Overall average population of each sample site showing the 10 most abundant bacterial orders across all sites. (**C**) The eight bacterial orders whose populations were determined to significantly differ between irrigated and nonirrigated sites, represented as counts per million reads. Error bars represent the standard deviation of triplicate data.

The online version of this article includes the following source data and figure supplement(s) for figure 1:

**Source data 1.** Read counts for data reported in *Figure 1*.

**Figure supplement 1.** Heatmap of the 50 most abundant bacterial orders across all samples (unclassified reads were removed from this analysis).

## Phenotypic, phylogenetic, and genomic analysis of the *P. fluorescens* field population

Taxonomic identifications using 16S rRNA amplicon analysis showed order-level changes to the field microbiome between sites (*Figure 1—figure supplement 1*) but were unable to accurately capture diversity within genera or species groups. Therefore, biologically relevant variation within the populations of genetically diverse species groups such as *P. fluorescens* is potentially overlooked. To investigate the diversity of the fluorescent pseudomonad population, we isolated 240 individual *Pseudomonas* strains from our pre- and post-irrigation field sites (*Supplementary file 1*). These strains were screened for multiple phenotypes including motility, protease production, fluorescence (siderophore production), and on-plate suppression of *S. scabies* using a cross-streak assay (*Figure 2—figure supplements 1 and 2*). Each phenotype was scored on an ordinal scale between 0 (no phenotype

observed) and 3 (strong phenotype). The cross-streak assay provided a rapid read-out of bacterial antagonism for both contact-dependent and diffusible mechanisms of growth inhibition. On-plate suppression of *S. scabies* was a surprisingly rare trait, with 79% of *Pseudomonas* isolates outcompeted by *S. scabies* in this assay. To determine whether this suppressive activity correlated with specific genetic loci, 69 isolates were selected for whole-genome sequencing, where almost half (32 strains) exhibited on-plate suppression of *S. scabies* and the remaining strains represented a diverse selection (based on phenotypic variation and 16S rRNA sequencing) of nonsuppressive strains. We hypothesized that a comparative analysis of a similar number of genomes from suppressive and nonsuppressive strains would identify those BGCs that play important roles in suppressive activity.

The phylogeny of the 69 sequenced strains was analyzed alongside various model pseudomonads, including representatives of the eight phylogenomic *P. fluorescens* groups defined by *Garrido-Sanz et al., 2016*. Our sequenced strain collection spans much of this characterized global phylogenetic diversity and contains representatives of at least five of the eight *P. fluorescens* phylogenomic groups (*Garrido-Sanz et al., 2016*), as well as strains belonging to the *Pseudomonas putida* and *Pseudomonas syringae* groups (*Figure 2—figure supplement 3*). This genetic heterogeneity was also reflected in the diverse specialized metabolome of these strains, as predicted by a detailed analysis of the BGCs encoded in their genomes. Each genome was subjected to antiSMASH 5.0 analysis (*Blin et al., 2019*), which was further refined by extensive manual annotation to improve the accuracy of predicted pathway products. This second annotation step was particularly important for BGCs that are atypically distributed across two distinct genomic loci (e.g., viscosin and pyoverdine). Our analysis was further expanded to include BGCs not identified by antiSMASH 5.0, including BGCs for hydrogen cyanide (HCN) (*Pessi and Haas, 2000*), microcin B17-like pathways (*Metelev et al., 2013*), and the auxin indole-3-acetic acid (IAA) (*McClerklin et al., 2018*; *Palm et al., 1989*). This was achieved by searching the genomes with a curated set of known *Pseudomonas* BGCs using MultiGeneBlast (*Medema et al., 2013*) (see Appendix 1 for further details). This manual annotation provided a level of resolution superior to that provided by automated cluster-searching algorithms alone and provided confidence that the majority of natural product biosynthetic potential had been identified. Within a given pathway type (e.g., nonribosomal peptide synthetases [NRPSs]), likely pathway products were assigned where possible (e.g., CLPs) or assigned a code when a conserved uncharacterized BGC was identified (e.g., NRPS 1). All BGCs were mapped to strain phylogeny (*Figure 2*).

Multiple BGCs were commonly found across the sequenced strains (*Figure 2*, *Supplementary file 1*), including BGCs predicted to make CLPs (*Raaijmakers et al., 2010*), arylpolyenes (*Cimermancic et al., 2014*), and HCN (*Pessi and Haas, 2000*). In addition to pyoverdine BGCs (*Cézard et al., 2015*) in almost all strains, numerous other siderophore BGCs were identified, including pathways predicted to make achromobactin (*Berti and Thomas, 2009*), ornicorrugatin (*Matthijs et al., 2008*), pyochelin-like molecules (*Patel and Walsh, 2001*; *Appendix 1—figure 1*), and a pseudomonine-like molecule (*Mercado-Blanco et al., 2001*). A variety of polyketide synthase (PKS), terpene, and NRPS BGCs with no characterized homologues were also identified (*Figure 2*). Furthermore, BGCs were identified that were predicted to make compounds related to microcin B17 (*Metelev et al., 2013*), fosfomycin-like antibiotics (*Kim et al., 2012*), lanthipeptides (*Repka et al., 2017*), safracin (*Velasco et al., 2005*), a carbapenem (*Coulthurst et al., 2005*), and an aminoglycoside (*Kudo and Eguchi, 2009*; *Appendix 1—figure 3*). Each of these natural product classes is predicted to have potent biological activity and some are rarely found in pseudomonads.

In addition to these potentially antibacterial and cytotoxic compounds, all genomes contain BGCs predicted to produce the plant auxin IAA, while 23 genomes contained genes for IAA catabolism (*Leveau and Gerards, 2008*). All 69 strains had at least one BGC for the production of the electron-transport cofactor pyrroloquinoline quinone (PQQ) (*Puehringer et al., 2008*; *Appendix 1—figure 4*), reported to function as a plant growth promoter (*Choi et al., 2008*). Surprisingly, BGCs for numerous well-characterized *Pseudomonas* specialized metabolites were not found, including phenazine, pyrrolnitrin, or 2,4-diacylphloroglucinol BGCs (*Gross and Loper, 2009*). In total, 787 gene clusters were identified that could be subdivided into 61 gene cluster families (*Figure 2*).

The *P. fluorescens* species group possesses a highly diverse array of nonessential accessory genes and gene clusters. These are often critical to the lifestyle of a given strain and can include motility determinants, proteases, secretion systems, polysaccharides, toxins, and metabolite catabolism pathways (*Mauchline et al., 2015*). These accessory genome loci were identified using MultiGeneBlast

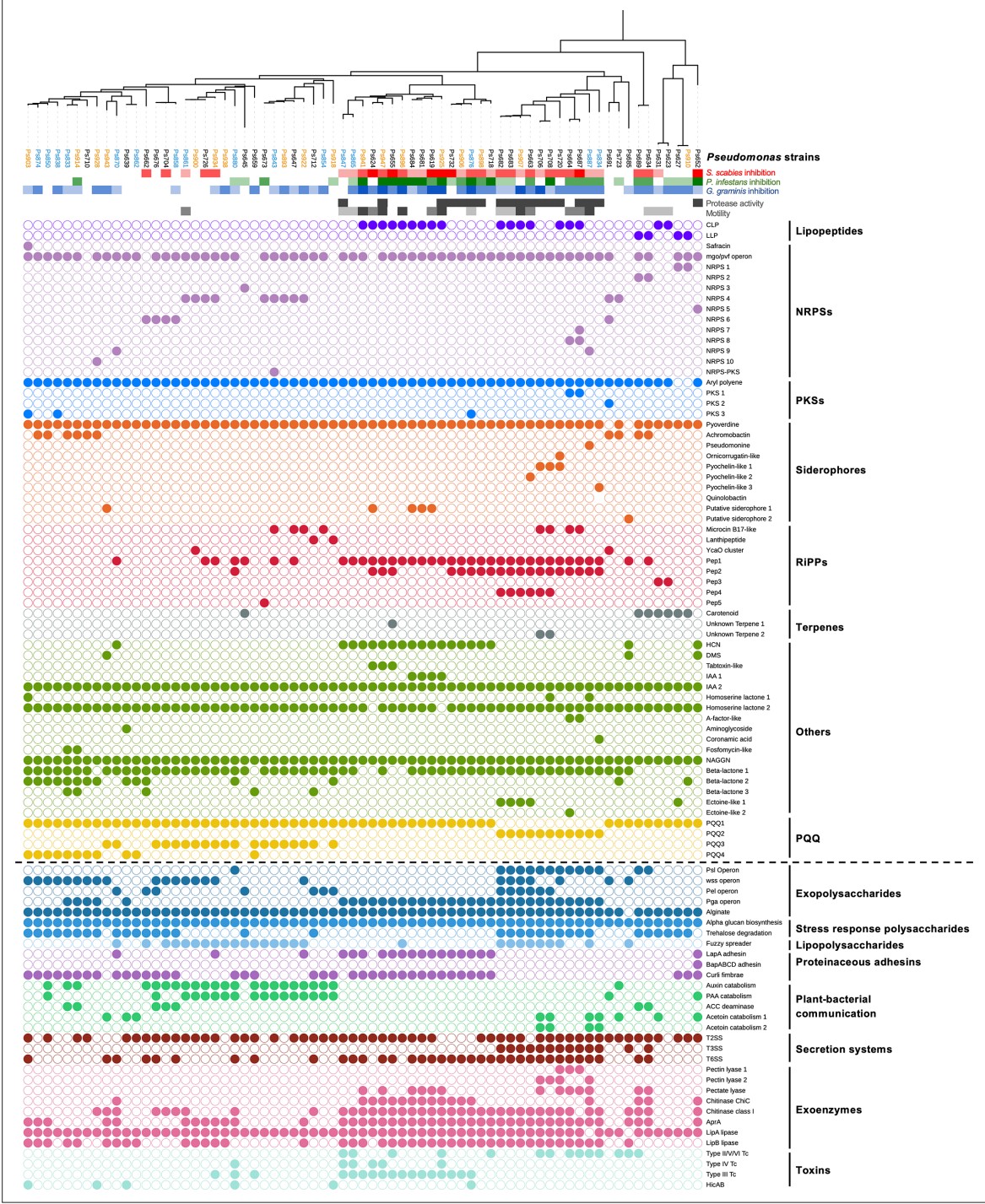

**Figure 2.** Comparison of phylogeny, *S. scabies* suppression (red color scales). *P. infestans* suppression (green color scale), *G. graminis* pv. *tritici* (take-all) suppression (blue color scale), phenotypes (gray color scales), natural product biosynthetic gene clusters (filled circles = presence of a gene or gene cluster), and the accessory genome (separated from biosynthetic gene clusters [BGCs] by a dotted line). In the phylogenetic tree of *Pseudomonas* strains, blue strains were collected from irrigated plots while orange strains were collected from unirrigated plots. All other strains were collected from the pre-irrigation plots. *Figure 2—figure supplement 3* shows the relationship between these strains and the *Pseudomonas* phylogenomic groups defined by *Garrido-Sanz et al., 2016*.

The online version of this article includes the following figure supplement(s) for figure 2:

**Figure supplement 1.** Representative images of *Pseudomonas* phenotypes.

**Figure supplement 2.** Representative images of on-plate suppressive activity of *Pseudomonas* isolates.

*Figure 2 continued on next page*

*Figure 2 continued*

**Figure supplement 3.** Maximum likelihood tree of full-length *gyrB* nucleotide sequences of *Pseudomonas* strains with representatives of the eight *Pseudomonas* phylogenomic groups defined by ***Garrido-Sanz et al., 2016***.

(details in Appendix 1), which revealed a high degree of genomic diversity across strains. Specialized metabolism BGCs and accessory genome loci were mapped to strain phylogeny (***Figure 2***), which indicated that for some loci (e.g., the *psl* operon, auxin catabolism, HCN biosynthesis) there is a close, but not absolute, relationship between phylogeny and the presence of a gene cluster.

## Correlation analysis identifies potential genetic determinants of *S. scabies* inhibition

We hypothesized that genes associated with suppression of *S. scabies* could be identified by a correlation analysis between *S. scabies* cross-streak inhibition and the presence of BGC families or accessory genes. We therefore calculated Pearson correlation coefficients for each BGC with *S. scabies* inhibition (***Figure 3—figure supplement 1***). The top 10 positively correlating genotypes and phenotypes (***Figure 3***) comprised four BGC families (Pep1, CLP, Pep2, and HCN) (***Appendix 1—figure 2***), four accessory genome loci (chitinase ChiC, protease AprA, chitinase class 1, and the Pga operon), and two phenotypes (motility and secreted protease production). The production of HCN and/or CLPs by *Pseudomonas* strains has been previously associated with the suppression of various plant pathogens including fungi (***Michelsen et al., 2015***; ***Zachow et al., 2015***) and oomycetes (***Hultberg et al., 2010***; ***Hunziker et al., 2015***), and can also contribute to insect killing (***Flury et al., 2017***; ***Jang et al., 2013***), but have not been linked to the suppression of bacteria. A variety of genotypes associated with plant-microbe interactions were moderately negatively correlated with suppression ($\rho < -0.3$), including BGCs for PQQ biosynthesis and catabolism of the plant auxins IAA and phenylacetic acid (PAA) (***Figure 3A***).

Interestingly, while certain BGC loci (e.g., CLP) positively correlated with both suppression and motility, this relationship was not seen for every locus (e.g., HCN correlates with suppression but is less strongly correlated with motility). Correlation does not equate to causation, especially considering the significant evolutionary association seen for some BGCs (***Figure 2***). The importance of correlating BGCs to *S. scabies* suppression was therefore investigated experimentally using a genetically tractable subset of suppressive isolates.

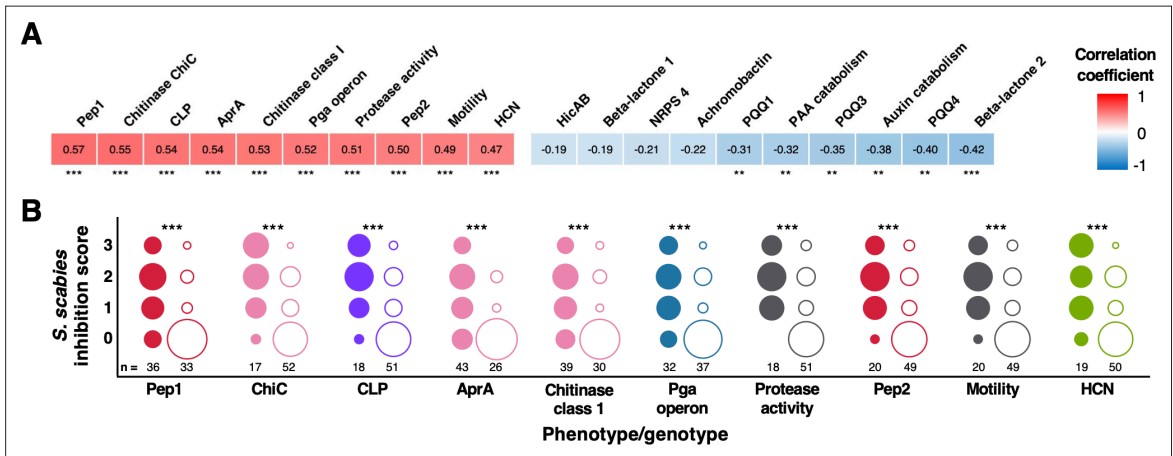

**Figure 3.** Correlation of biosynthetic gene clusters (BGCs) and accessory genome loci with *S. scabies* inhibition. (**A**) Heatmap showing the 10 genotypes and phenotypes that correlated most strongly (positively and negatively) with on-plate suppression of *S. scabies*. Stars represent the statistical significance of a correlation using a two-tailed Mann–Whitney test (\*p<0.05, \*\*p<0.01, \*\*\*p<0.001). (**B**) Distributions of *S. scabies* suppressive activity for top 10 positive correlations. Circles are stacked from no (0) to high (3) inhibition, where filled and empty circles represent strains with and without a given genotype/phenotype, respectively. The number of strains (total = 69) in each class is listed, and the area of a circle specifies the proportion of strains with given suppressive activity.

The online version of this article includes the following figure supplement(s) for figure 3:

**Figure supplement 1.** Heatmap showing Pearson correlation coefficients of phenotypes and genotypes across the sequenced *Pseudomonas* isolates.

## Production and role of CLPs in the suppression of *S. scabies*

The strong positive correlation between putative CLP gene clusters and *S. scabies* suppression prompted us to investigate whether CLPs play a role in suppressive activity. *Pseudomonas* CLPs have previously been associated with a wide array of functions, including fungal growth inhibition, plant colonization, and promotion of swarming motility (*Alsohim et al., 2014*; *Raaijmakers et al., 2010*), although there are no reports of *Pseudomonas* CLPs functioning as inhibitors of streptomycete growth. However, prior work has shown that surfactin, a CLP from *Bacillus subtilis,* inhibits *Streptomyces coelicolor* aerial hyphae development (*Straight et al., 2006*), while iturin A, a CLP from *Bacillus* sp. sunhua, inhibits *S. scabies* development (*Han et al., 2005*).

To determine the identity of each CLP, we combined bioinformatic predictions of the NRPS products (*Blin et al., 2019*) with experimental identification using liquid chromatography–tandem mass spectrometry (LC-MS/MS). In every strain that contained a CLP BGC, a molecule with an expected mass and MS/MS fragmentation pattern was identified (*Figure 4*, *Figure 4—figure supplements 1–7*). These data showed that *P. fluorescens* strains from a single field have the collective capacity to make viscosin ($m/z$ 1126.69, identical retention time to a viscosin standard) (*de Bruijn et al., 2007*), a viscosin isomer ($m/z$ 1126.69, different retention time to viscosin standard) (*Figure 4—figure supplements 1 and 2*), as well as compounds with BGCs, exact masses, and MS/MS fragmentation consistent with tensin ($m/z$ 1409.85, *Figure 4—figure supplement 3*; *Nielsen et al., 2000*), anikasin ($m/z$ 1354.81, *Figure 4—figure supplement 4*; *Götze et al., 2017*), and putisolvin II ($m/z$ 1394.85, *Figure 4—figure supplement 5*; *Kuiper et al., 2004*). In addition, an array of related metabolites were observed that differed by 14 or 28 Da, which is characteristic of different lipid chain lengths. This analysis also proved that the linear lipopeptides syringafactin A ($m/z$ 1082.74) and cichofactin ($m/z$ 555.38, $[M + 2 H]^{2+}$) were made by strains harboring BGCs predicted to make these phytotoxins (*Götze et al., 2019*; *Pauwelyn et al., 2013*; *Figure 4—figure supplements 6 and 7*). The metabolic capacity of all strains was mapped using mass spectral networking (*Aron et al., 2020*; *Wang et al., 2016*), which showed that CLPs were strongly associated with strains that inhibit *S. scabies* (*Figure 4*).

To assess the potential role of CLPs in mediating the interaction between *P. fluorescens* and *S. scabies*, an NRPS gene predicted to be involved in the biosynthesis of a viscosin-like molecule in Ps682 was deleted by allelic replacement (*Figure 5A*). The resulting Ps682 *Δvisc* strain was unable to make the viscosin-like molecule ($m/z$ 1126.69, *Figure 5B*) or to undergo swarming motility (*Figure 5—figure supplement 1*). This is in agreement with earlier work on the role of viscosin in the motility of *P. fluorescens* SBW25 (*Alsohim et al., 2014*) and the observation that possession of a CLP BGC was the genotype that most strongly correlated with motility ($\rho$ = 0.65, *Figure 3—figure supplement 1*). A cross-streak assay with *S. scabies* revealed an active role for this CLP in on-plate *S. scabies* inhibition (*Figure 5D*). Wild-type (WT) Ps682 appeared to specifically colonize the *S. scabies* streak, whereas Ps682 *Δvisc* was unable to restrict *S. scabies* growth. Alternatively, it was possible that this instead could reflect diffusible inhibition of *Streptomyces* development by WT Ps682, leading to a 'bald' *S. scabies* phenotype (*Flärdh and Buttner, 2009*). To distinguish between these possible inhibition modes, a constitutively expressed *lux* operon was integrated into the chromosomal *att::*Tn*7* site (K.-H. *Choi et al., 2005*) of Ps682 to visualize this interaction by bioluminescence. This clearly showed viscosin-dependent *Pseudomonas* colonization of the *Streptomyces* streaks (*Figure 5D*).

To quantitatively assess the antagonistic effect of the Ps682 CLP, it was purified and structurally characterized using MS/MS (*Figure 5—figure supplement 2*) and nuclear magnetic resonance (NMR) spectroscopy ($^{1}$H, $^{13}$C, COSY HSQC, TOCSY, HMBC, Figure 5—figure supplements 3–13, *Supplementary file 2D*). NMR analysis revealed that the molecule has an identical amino acid composition to viscosin (3-hydroxydecanoic acid-Leu1-Glu2-Thr3-Val4-Leu5-Ser6-Leu7-Ser8-Ile9, *Figure 5B*), which was fully supported by detailed high-resolution MS (calculated viscosin $[M + H]^{+}$ = 1126.6970, observed $[M + H]^{+}$ = 1126.6964) and MS/MS fragmentation data (*Figure 5—figure supplement 2*). The LC retention time of this CLP is different to viscosin, but is almost identical to WLIP (*Figure 5—figure supplement 2C*), which is a viscosin isomer that has a D-Leu5 residue instead of L-Leu5 (*Rokni-Zadeh et al., 2012*). However, comparison of NMR data in DMF-d7 revealed some minor shift differences between published WLIP spectra (*Rokni-Zadeh et al., 2012*) and the Ps682 CLP, such as the γ-CH$_2$ group of Glu2 (WLIP = $\delta_H$ 2.54 ppm, $\delta_C$ 30.3 ppm; Ps682 CLP = $\delta_H$ 2.24 ppm, $\delta_C$ 34.8 ppm). Therefore, we could not conclusively confirm the absolute configuration of the Ps682 CLP and thus named it viscosin I (for viscosin Isomer). A disk diffusion assay of purified viscosin I with *S. scabies* (*Figure 5C*)

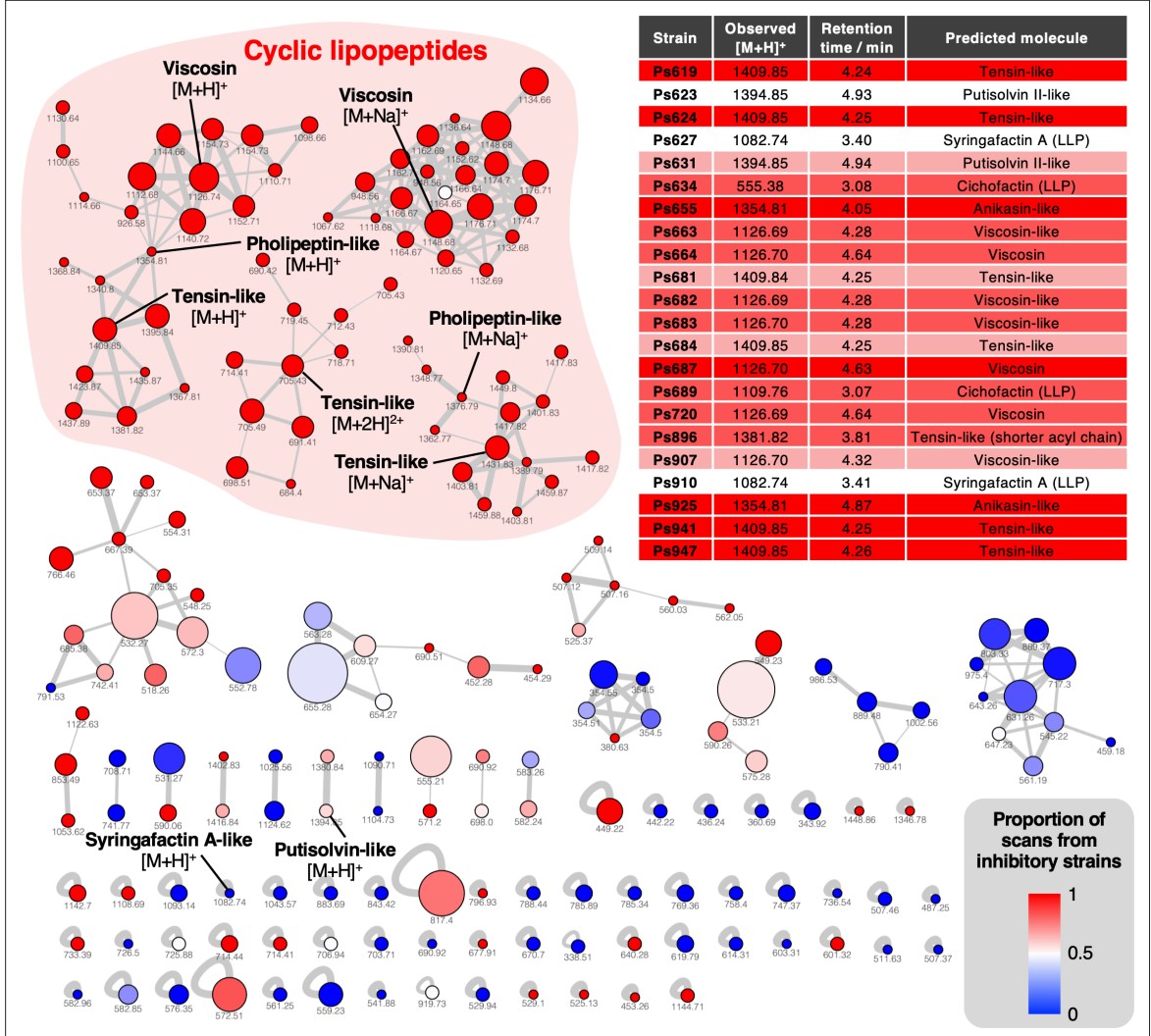

| Strain | Observed [M+H]+ | Retention time / min | Predicted molecule |
|---|---|---|---|
| Ps619 | 1409.85 | 4.24 | Tensin-like |
| Ps623 | 1394.85 | 4.93 | Putisolvin II-like |
| Ps624 | 1409.85 | 4.25 | Tensin-like |
| Ps627 | 1082.74 | 3.40 | Syringafactin A (LLP) |
| Ps631 | 1394.85 | 4.94 | Putisolvin II-like |
| Ps634 | 555.38 | 3.08 | Cichofactin (LLP) |
| Ps655 | 1354.81 | 4.05 | Anikasin-like |
| Ps663 | 1126.69 | 4.28 | Viscosin-like |
| Ps664 | 1126.70 | 4.64 | Viscosin |
| Ps681 | 1409.84 | 4.25 | Tensin-like |
| Ps682 | 1126.69 | 4.28 | Viscosin-like |
| Ps683 | 1126.70 | 4.28 | Viscosin-like |
| Ps684 | 1409.85 | 4.25 | Tensin-like |
| Ps687 | 1126.70 | 4.63 | Viscosin |
| Ps689 | 1109.76 | 3.07 | Cichofactin (LLP) |
| Ps720 | 1126.69 | 4.64 | Viscosin |
| Ps896 | 1381.82 | 3.81 | Tensin-like (shorter acyl chain) |
| Ps907 | 1126.70 | 4.32 | Viscosin-like |
| Ps910 | 1082.74 | 3.41 | Syringafactin A (LLP) |
| Ps925 | 1354.81 | 4.87 | Anikasin-like |
| Ps941 | 1409.85 | 4.25 | Tensin-like |
| Ps947 | 1409.85 | 4.26 | Tensin-like |

**Figure 4.** Mass spectral networking analysis of liquid chromatography–tandem mass spectrometry (LC-MS/MS) data from the *Pseudomonas* strains used in this study. Node area is proportional to the number of distinct strains where MS/MS data were acquired for a given metabolite. Node color reflects the proportion of MS/MS scans for a given node that come from strains with a *S. scabies* inhibition score ≥1. Nodes are labeled with the corresponding parent masses and nodes that relate to lipopeptides are labeled (multiple networks arise from differential fragmentation of [M + H]+, [M + 2H]2+, and [M + Na]+ ions). Line thickness is proportional to the cosine similarity score calculated by Global Natural Product Social Molecular Networking (GNPS) (*Aron et al., 2020*). The table shows production of lipopeptides by strains containing lipopeptide biosynthetic gene clusters (BGCs). Color coding reflects level of *S. scabies* inhibition by each strain with same scale as *Figure 2* (LLP: linear lipopeptide; all others are cyclic lipopeptides [CLPs]).

The online version of this article includes the following figure supplement(s) for figure 4:

**Figure supplement 1.** Liquid chromatography–tandem mass spectrometry (LC-MS/MS) spectra showing that some strains produce viscosin (identical retention time and MS/MS fragmentation to viscosin produced by *P. fluorescens* SBW25; green boxes) and some produce an isomer with a distinct retention time and MS/MS fragmentation pattern (yellow boxes).

**Figure supplement 2.** Comparison of viscosin (**A**) and viscosin-like (**B**) biosynthetic gene clusters (BGCs).

**Figure supplement 3.** Genetic and liquid chromatography–tandem mass spectrometry (LC-MS/MS) analysis of tensin-like compounds.

**Figure supplement 4.** Characterization of an anikasin-like lipopeptide from strain Ps655.

**Figure supplement 5.** Characterization of a putisolvin II-like lipopeptide from strain Ps623.

**Figure supplement 6.** Characterization of a syringafactin-like lipopeptide from strain Ps627.

**Figure supplement 7.** Characterization of a cichofactin-like lipopeptide from strain Ps689.

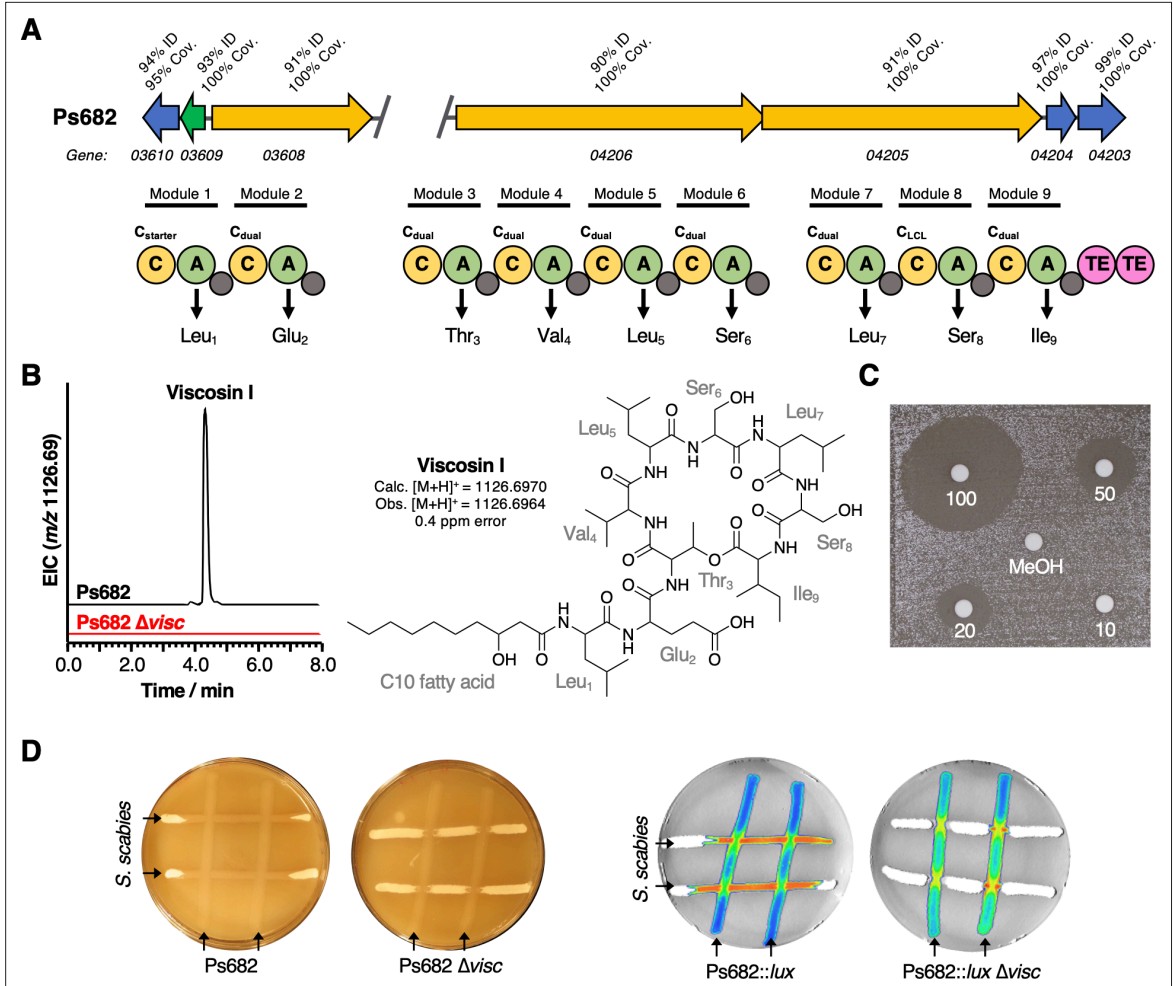

**Figure 5.** The role of the Ps682 cyclic lipopeptide (CLP) biosynthetic gene cluster (BGC) in *S. scabies* suppression. (**A**) BGC displaying identity/coverage scores in comparison to the viscosin BGC in *P. fluorescens* SBW25. Genes encoding regulatory proteins are green, transporter genes are blue, and nonribosomal peptide synthetase (NRPS) genes are yellow. The NRPS organization is shown, where C = condensation domain, A = adenylation domain, TE = thioesterase domain, and gray circles are peptidyl carrier protein domains. Amino acids incorporated by each module are displayed, along with predicted condensation domain specificity. (**B**) Liquid chromatography–mass spectrometry (LC-MS) analysis of viscosin I production in the WT strain and a mutant (Ps682 Δ*visc*) with an in-frame deletion of NRPS gene 04206 (EIC = extracted ion chromatogram). Nuclear magnetic resonance (NMR) and MS/MS data for viscosin I are shown in *Figure 5—figure supplements 2–13*. (**C**) Disk diffusion assay of viscosin I against *S. scabies*. Concentrations are indicated (μg/mL), alongside a methanol control. (**D**) On-plate *S. scabies* suppression activity of Ps682 alongside Ps682 Δ*visc* shown as cross-streaks using strains with and without the *lux* operon. Bioluminescence was detected using a NightOWL camera (Berthold Technologies).

The online version of this article includes the following figure supplement(s) for figure 5:

**Figure supplement 1.** Swarming motility of wild-type Ps682 and Ps682 Δ*visc*.

**Figure supplement 2.** Detailed mass spectrometry (MS) analysis of viscosin-like molecule produced by Ps682.

**Figure supplement 3.** Atom numbering for viscosin I nuclear magnetic resonance (NMR) annotation.

**Figure supplement 4.** ¹H nuclear magnetic resonance (NMR) spectrum of viscosin I (600 MHz, DMF-d7, 298 K).

**Figure supplement 5.** ¹³C nuclear magnetic resonance (NMR) spectrum (DEPT135) of viscosin I (600 MHz, DMF-d7, 298 K).

**Figure supplement 6.** 2D COSY nuclear magnetic resonance (NMR) spectrum of viscosin I (600 MHz, DMF-d7, 298 K).

**Figure supplement 7.** Expanded regions of 2D COSY nuclear magnetic resonance (NMR) spectrum of viscosin I (600 MHz, DMF-d7, 298 K).

**Figure supplement 8.** 2D HSQC-EDITED nuclear magnetic resonance (NMR) spectrum of viscosin I (600 MHz, DMF-d7, 298 K).

**Figure supplement 9.** Expansion of (**A**) α-C, (**B**) side chain, and (**C**) Me-group regions of 2D HSQC-EDITED nuclear magnetic resonance (NMR) spectrum of viscosin I (600 MHz, DMF-d7, 298 K).

**Figure supplement 10.** 2D TOCSY nuclear magnetic resonance (NMR) spectrum of viscosin I (600 MHz, DMF-d7, 298 K).

**Figure supplement 11.** 2D HMBC nuclear magnetic resonance (NMR) spectrum of viscosin I (600 MHz, DMF-d7, 298 K).

*Figure 5 continued on next page*

*Figure 5 continued*

**Figure supplement 12.** 2D HSQC-TOCSY nuclear magnetic resonance (NMR) spectrum of viscosin I (600 MHz, DMF-d7, 298 K).

**Figure supplement 13.** High-field region of 2D HSQC-TOCSY nuclear magnetic resonance (NMR) spectrum of viscosin I (600 MHz, DMF-d7, 298 K).

**Figure supplement 14.** Time course of viscosin I disk diffusion assays against *S. scabies* 87–22 on Instant Potato Medium (IPM).

demonstrated that it directly inhibited *S. scabies* growth with a minimum inhibitory concentration of approximately 20 µg/mL. Long-term growth of *S. scabies* in the presence of viscosin I (*Figure 5—figure supplement 14*) indicated that the inhibition of *S. scabies* is temporary and growth partially resumes after several days. These data show that in addition to its role as a surfactant viscosin I functions by inhibiting the growth rate of *S. scabies*, consistent with the on-plate data for Ps682 Δ*visc*.

## HCN and CLP production both contribute to on-plate *S. scabies* inhibition

Pan-genome analysis showed that HCN production was predicted for a significant number of suppressive strains ($\rho$ = 0.47, *Figure 3—figure supplement 1*), where 17 of the 19 strains containing HCN gene clusters were inhibitory towards *S. scabies* (*Figure 2*). The HCN pathway is encoded by the *hcnABC* gene cluster (*Appendix 1—figure 2*) and has previously been associated with insect and fungal pathogen inhibition in other *Pseudomonas* strains (*Flury et al., 2017*; *Hunziker et al., 2015*; *Siddiqui et al., 2006*). HCN is toxic to a wide variety of organisms, but not to *Pseudomonas* owing to their branched aerobic respiratory chain that has at least five terminal oxidases, including a cyanide-insensitive oxidase (*Comolli and Donohue, 2002*; *Ugidos et al., 2008*). We confirmed that nearly every strain with the *hcnABC* gene cluster produced HCN (18 out of 19) using the Feigl–Anger colorimetric detection reagent (*Feigl and Anger, 1966*; *Supplementary file 1*) and used this assay to identify HCN producers across the original collection of 240 *Pseudomonas* strains. This wider analysis showed that HCN production strongly correlated with *S. scabies* inhibition ($\rho$ = 0.52, *Figure 6—figure supplement 1*), in accordance with our analysis of the sequenced strains.

To examine the role of HCN in *S. scabies* suppression and whether it exhibited a synergistic effect with CLP production, Ps619 was investigated as this strain produces both HCN and a tensin-like CLP (*Figures 2 and 6A*). A tensin BGC has not previously been reported, but the predicted amino acid specificity, mass (*Figure 6A*), and MS/MS fragmentation (*Figure 4—figure supplement 3*) indicated that seven isolates produce tensin-like CLPs (*Figure 4*). The *hcn* and *ten* gene clusters were inactivated by in-frame deletions to generate single and double mutants of Ps619, and the resulting Δ*hcn*, Δ*ten,* and Δ*hcn*Δ*ten* mutants were subjected to cross-streak assays (*Figure 6B*). A comparison of WT, single, and double mutants showed that HCN inhibits *S. scabies* growth and development across the entire plate, while tensin is important for *Pseudomonas* motility and helps the *Pseudomonas* to grow competitively at the cross-streak interface. Furthermore, this suppressive effect is additive: the Ps619 Δ*hcn* and Δ*ten* single mutants both retained some inhibitory activity towards *S. scabies*, whereas the Ps619 Δ*hcn*Δ*ten* double mutant could not inhibit *S. scabies*.

In drier growth conditions expected to favor streptomycete growth and limit motility, the role of tensin-mediated motility was abrogated, yet tensin and HCN still possessed an additive inhibitory effect at the microbial interface (*Figure 6—figure supplement 2A*). Notably, Ps619 Δ*hcn* was able to induce a developmental defect in *S. scabies* at the microbial interface that was not present in Ps619 Δ*ten* or Ps619 Δ*hcn*Δ*ten*, showing that the tensin-like CLP induces a developmental defect in *S. scabies* that is independent of *Pseudomonas* motility, comparable to the inhibitory effect of isolated viscosin I. This analysis also clearly showed that at areas distant from the bacterial interaction *S. scabies* grew more vigorously when cultured with Δ*hcn* strains, consistent with the volatility of HCN enabling a long-range inhibitory effect. A similar volatile effect was seen when Ps619 strains were separated from *S. scabies* by a barrier, where only those strains producing HCN inhibited growth and development (*Figure 6—figure supplement 2B*).

To further probe how tensin and HCN affected the interaction between Ps619 strains and *S. scabies*, the interfacial regions of cross-streaks were imaged using cryo-scanning electron microscopy (cryo-SEM). WT Ps619 was able to colonize the *S. scabies* streak, meaning that the interfacial region imaged was further from the cross-streak intersection than all other co-cultures (*Figure 6C*). Here, Ps619 inhibited *S. scabies* development, which appears as a mixture of deformed aerial hyphae and

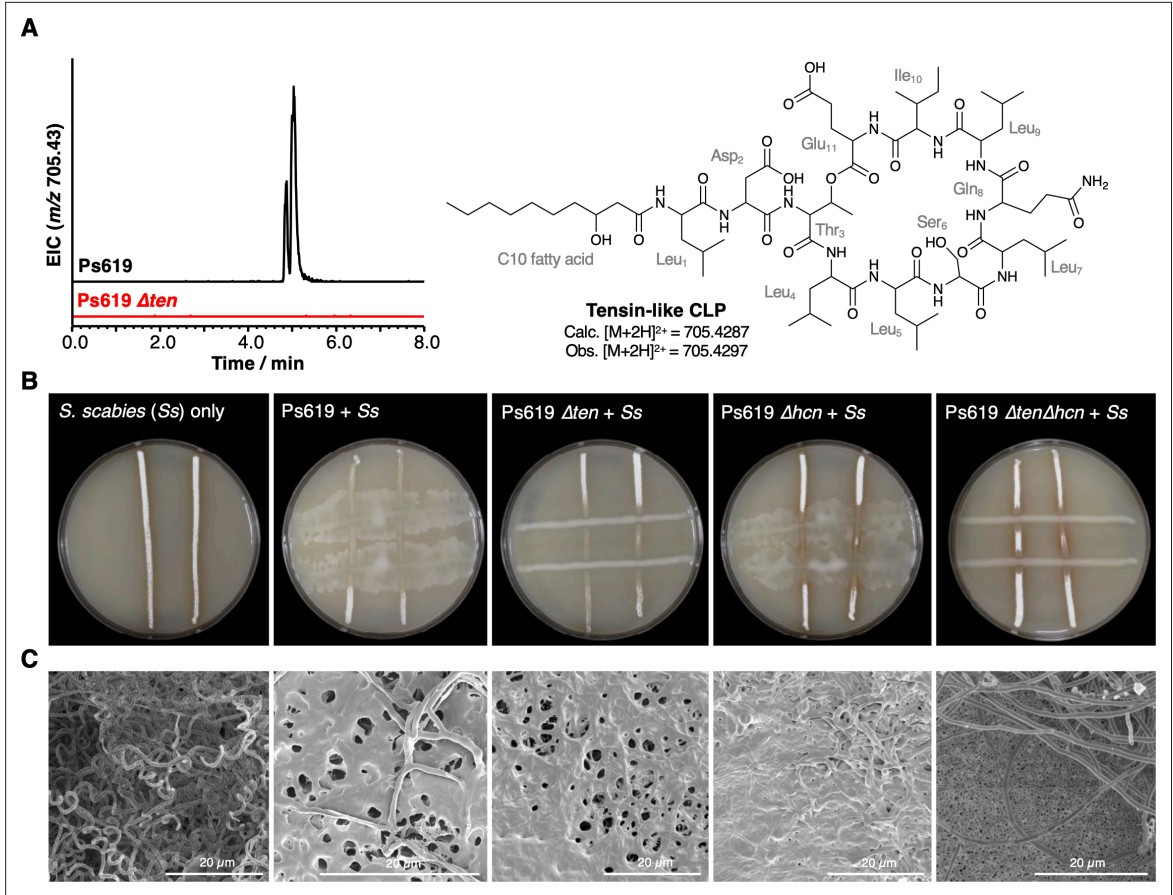

**Figure 6.** The role of the Ps619 cyclic lipopeptide (CLP) and hydrogen cyanide (HCN) gene clusters in *S. scabies* suppression. (**A**) Predicted structure of the tensin-like molecule and liquid chromatography–tandem mass spectrometry (LC-MS) analysis of CLP production in wild-type (WT) Ps619 and a mutant (Ps619 *Δten*) with an in-frame deletion of nonribosomal peptide synthetase (NRPS) gene 02963 (see *Figure 4—figure supplement 3* for biosynthetic gene cluster [BGC] information). (**B**) Cross-streak assays of Ps619 and associated mutants with *S. scabies*. See *Figure 6—figure supplement 2A* for assays with drier plates. (**C**) Cryo-scanning electron microscopy (Cryo-SEM) images of the interfacial region between the Ps619 strains and *S. scabies*. The order of images is identical to the cross-streaks in panel (**B**).

The online version of this article includes the following figure supplement(s) for figure 6:

**Figure supplement 1.** Pearson correlation coefficients for phenotypes across both sequenced and unsequenced isolates.

**Figure supplement 2.** Effect of Ps619 on *S. scabies* growth and development.

**Figure supplement 3.** Further cryo-scanning electron microscopy (cryo-SEM) images of the interfacial region between the Ps619 strains and *S. scabies* (*Ss*) showing that a well-defined boundary only exists with the Ps619 *ΔtenΔhcn* strain (bottom right).

vegetative growth reminiscent of a 'bald' phenotype (*Tschowri et al., 2014*). Cryo-SEM indicated that both the Ps619 *Δhcn* and *Δten* mutants induced a similar partially bald phenotype in *S. scabies*, but the *ΔhcnΔten* double mutant was unable to trigger the same developmental defect as *S. scabies* could develop aerial mycelia close to the microbial interface (*Figure 6C*, *Figure 6—figure supplement 3*). This appears as a clear boundary between Ps619 *ΔhcnΔten* (single cells in background, bottom right panel of *Figure 6C*) and *S. scabies* (hyphae in the foreground). The volatile HCN can inhibit growth and development at a distance, whereas CLP inhibition of development only occurs close to the microbial interface. Both inhibitory mechanisms enable Ps619 to obtain a competitive advantage at the microbe-microbe interface (*Figure 6C*, *Figure 6—figure supplement 3*), while the CLP also functions as a surfactant enabling Ps619 motility, promoting *Pseudomonas* invasion of the *Streptomyces* cross-streak.

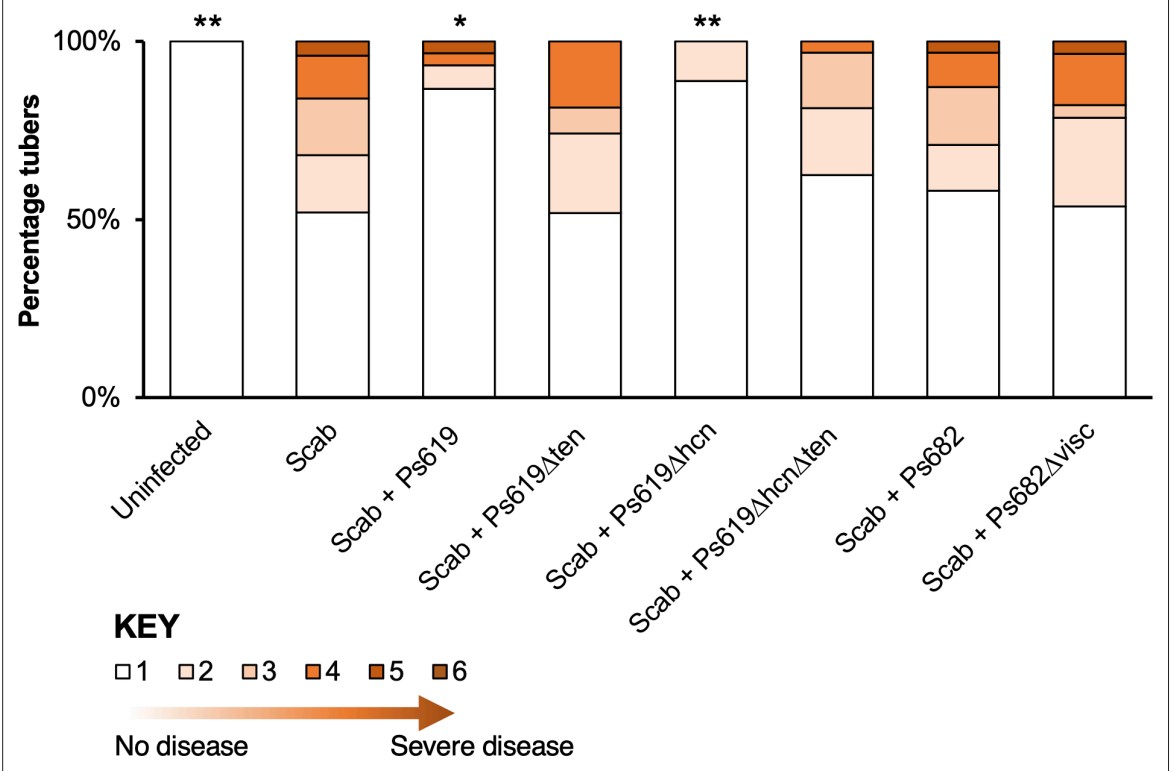

**Figure 7.** Potato scab biocontrol assay. The bar chart shows the percentage of diseased tubers following infection with *S. scabies* ('Scab') along with treatment by Ps619, Ps682, and associated mutants. Tubers were scored using a disease severity index from 1 to 6 according to the method of *Andrade et al., 2019*. Statistical analyses were calculated by taking into account the average disease index of each plant (n = 4). p-Values were calculated using Dunnett's multiple comparison test, and asterisks indicate *p<0.05, **0.01 as compared to scab treatment only. Results of a repeat biocontrol experiment and further statistical statistics are shown in *Figure 7—figure supplement 1*.

The online version of this article includes the following figure supplement(s) for figure 7:

**Figure supplement 1.** Results from set 2 of the potato scab biocontrol assay (set 1 data shown in *Figure 7*) alongside statistical summaries of the two sets of experiments.

## Tensin is a key determinant of in planta inhibition of potato scab

To examine the in planta biocontrol properties of Ps619 and Ps682, and to determine the contribution of HCN and CLPs to activity, potato scab suppression assays were carried out in glasshouse trials. Maris Piper potatoes were infected with *S. scabies* 87-22 and scored for disease severity after 16 weeks using the method of *Andrade et al., 2019*. A subset of plants was also treated with *Pseudomonas* spp. and associated BGC mutants. Ps619 conferred significant protection against potato scab, where disease severity was reduced to levels similar to uninfected control plants (*Figure 7*). This suppressive ability was lost for Ps619 Δ*ten* and Ps619 Δ*hcn*Δ*ten*, resulting in disease severity similar to scab-infected tubers. In contrast, Ps619 Δ*hcn* was just as effective as WT Ps619 at suppressing potato scab, which differed from the on-plate results for HCN. The significance of these results was supported by an independent in planta biocontrol experiment, where equivalent results were obtained for each strain (*Figure 7—figure supplement 1*). This result indicates that tensin plays an important role in the biocontrol of potato scab. In contrast to its on-plate suppressive activity, potato scab assays showed no significant antagonistic activity for Ps682 against *S. scabies* infection. Unfortunately, this meant that the role of viscosin I could not be determined in planta.

## A subset of *P. fluorescens* strains are generalist pathogen suppressors

To determine whether the strains and metabolites we identified have suppressive activity towards a range of plant pathogens, we investigated the ability of the potato field strain collection to suppress the growth of *Phytophthora infestans*, the oomycete that causes potato blight (*Nowicki et al., 2012*), and *Gaeumannomyces graminis* var. *tritici*, the fungus that causes take-all disease of cereal

crops (*Mauchline et al., 2015*). These assays revealed strong congruence between the genotypes that correlated with suppression of each pathogen (*Appendix 2—figure 1A*). HCN and CLPs have both previously been identified as inhibitors of oomycete and fungal growth (*Hunziker et al., 2015*; *Michelsen et al., 2015*). To assess whether these natural products are critical for inhibition of *P. infestans* and *G. graminis* by Ps619 and Ps682, the HCN/CLP mutants were tested for inhibitory activity (*Appendix 2—figure 1*). Surprisingly, neither HCN or tensin were required for Ps619 inhibition of either *G. graminis* or *P. infestans* (*Appendix 2—figure 1*), indicating the production of at least one other secreted inhibitory factor. In contrast, inactivation of the viscosin I pathway in Ps682 abolished activity towards both pathogens (*Appendix 2—figure 1D*). These data indicate that a subset of pseudomonads can function as generalist pathogen suppressors, possessing multiple growth inhibition mechanisms (e.g., Ps619) and/or by producing molecules with broad bioactivity (e.g., Ps682).

Multiple other genome loci are strongly correlated with pathogen suppression (*Figure 3A*, *Appendix 2—figure 1A*), including chitinases (*Folders et al., 2001*) and the extracellular metalloprotease AprA (*Laarman et al., 2012*). Phenotypically, extracellular protease activity also positively correlates with suppression. The BGC that correlated most strongly with *S. scabies* suppression was Pep1 ('Peptide 1'), while the related Pep2 also correlated strongly (*Figure 3A*). These were identified by antiSMASH as putative 'bacteriocin' BGCs and encode short DUF2282 peptides alongside DUF692 and DUF2063 proteins (*Appendix 1—figure 2*). The DUF692 protein family includes dioxygenases involved in methanobactin (*Kenney et al., 2018*) and 3-thiaglutamate (*Ting et al., 2019*) biosynthesis. Other studies indicate that DUF692 and DUF2063 proteins may be involved in heavy metal and/or oxidative stress responses (*Clark et al., 2014*; *Price et al., 2018*; *Sarkisova et al., 2014*). Further work is required to determine the significance of both the Pep BGCs and the accessory genome loci for pathogen inhibition.

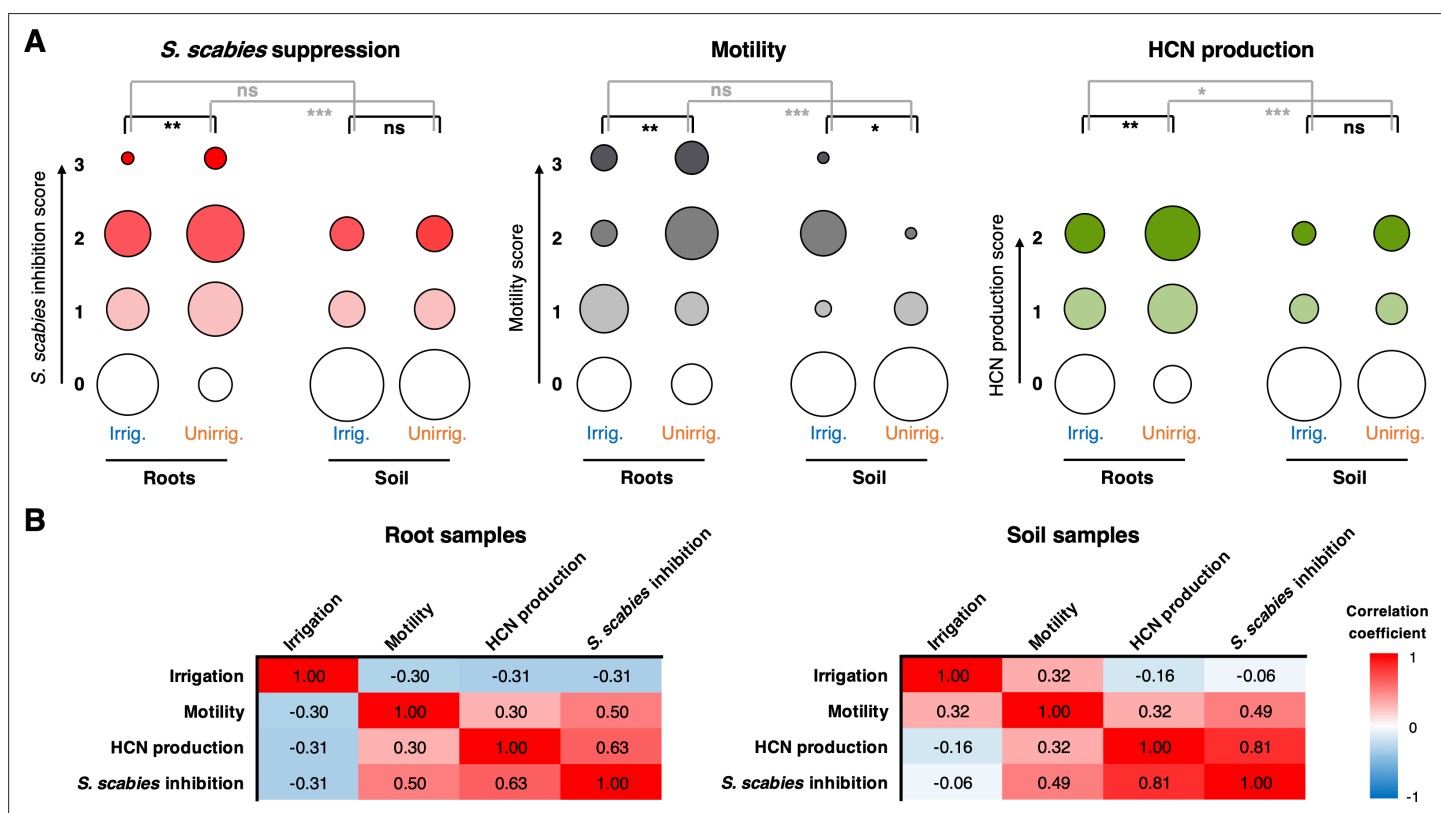

**Figure 8.** The effect of irrigation and environment (soil versus root) on the *P. fluorescens* population. (**A**) Plots showing the proportion of strains exhibiting a particular phenotype from each environment (n = 48 for each condition). Hydrogen cyanide (HCN) production was scored on a scale of 0–2 based on a qualitative assessment of the color change in the Feigl–Anger assay. The size of each circle is proportional to the number of strains with a given phenotypic score. Statistical comparisons were carried out using two-tailed Mann–Whitney tests where ns (not significant) = p≥0.05, *p<0.05, ** p<0.01, ***p<0.001. (**B**) Pearson correlation scores for phenotypes from strains isolated from roots (n = 96) and soil (n = 96).

## The effect of irrigation on the soil *Pseudomonas* population

Irrigation is currently the only effective way to control potato scab, so we hypothesized that this may lead to an increase in the number of inhibitory bacteria associated with the soil and/or tuber, especially as the *Pseudomonadales* population moderately increased in irrigated soil (*Figure 1C*). However, a greater number of strongly suppressive strains (inhibition score ≥2) were isolated from nonirrigated sites (7/60 strains) than from irrigated sites (1/60 strains). A similar pattern was observed for strongly motile (score ≥2) strains (six nonirrigated versus two irrigated). Analysis of the BGCs in our sequenced strains revealed a similar result, where 5/18 unirrigated strains contained CLP BGCs versus 0/16 irrigated strains. This counterintuitive observation led us to hypothesize that irrigation enables nonmotile, nonsuppressive bacteria to survive and colonize plant roots, whereas highly motile bacteria that produce multiple biological weapons can more effectively colonize plants in drier, more 'hostile' conditions.

To test these hypotheses, we sampled irrigated and unirrigated sites in a neighboring field 2 years after the first sampling event. 48 strains were isolated from bulk soil and the rhizospheres of tuber-forming potato plants, with and without irrigation, providing a total of 192 *P. fluorescens* strains (*Supplementary file 1*). These strains were scored for motility, HCN production, and *S. scabies* suppression (*Figure 8A*). Our results were in strong agreement with the first sample set, including strong positive correlations between *S. scabies* inhibition, motility, and HCN production (*Figure 8B*). A negative correlation was observed between irrigation and *S. scabies* suppression on the plant roots, but not in the surrounding soil. This appeared to be driven primarily by differences in the unirrigated samples, where a substantially greater proportion of suppressive isolates were associated with roots than with the surrounding soil. We observed a strong positive correlation between motility and root association for unirrigated samples, while the reverse was true for irrigated plants (*Figure 8A*). This effect of irrigation on the distribution of motile bacteria was striking – in dry plants, the motile population was almost entirely associated with roots, while in irrigated plants a comparable proportion of motile bacteria were found in the soil and roots (*Figure 8A*).

This analysis therefore supports the root colonization hypothesis, where a lack of irrigation leads to a more specialized pseudomonad population colonizing the root. Upon irrigation, the difference between the bulk soil and root pseudomonad populations is much less significant. The mechanism for this population change is not yet defined, and these changes are counterintuitive in relation to the suppression of potato scab upon irrigation, given there is a drop in suppressive strains colonizing the potato root following irrigation. Irrigation did lead to moderately more motile pseudomonads in bulk soil versus unirrigated conditions, but this was not associated with more suppressive strains or HCN producers (*Figure 8A and B*). The mechanism and significance of this irrigation effect require further investigation. It is possible that a protective microbiome in irrigated conditions actually contains a mixture of *S. scabies*-suppressive 'biocontrol' *Pseudomonas* strains alongside other nonmotile pseudomonads that interact with the plant in important ways due to traits usually absent from the 'biocontrol' strains, such as their ability to produce PQQ and catabolize auxins (*Choi et al., 2008*; *Leveau and Gerards, 2008*). Profound irrigation-associated changes in antibiotic-producing *Pseudomonas* populations have previously been observed for the wheat rhizosphere (*Mavrodi et al., 2018*; *Mavrodi et al., 2012*).

## Discussion

Prior studies on the suppression of potato scab have indicated a potential biocontrol role for *Pseudomonas* bacteria (*Arseneault et al., 2015*; *Arseneault et al., 2013*; *Elphinstone et al., 2009*; *Rosenzweig et al., 2012*). Fluorescent pseudomonads form multiple beneficial relationships with plants, including growth promotion and biocontrol (*Haas and Défago, 2005*; *Zamioudis et al., 2013*). However, there is limited understanding of the genetic factors that are critical for such activity, and little is known about the diversity of the *P. fluorescens* species group within a given agricultural field or how this population is shaped by environmental changes. In this study, we integrated genomics, metabolomics, phenotypic analysis, molecular biology, and in planta assays to identify the genetic determinants of *Pseudomonas* antagonism towards *S. scabies*. This population-level approach shows that the *P. fluorescens* population in a single field is highly complex, heterogeneous, and dynamic (*Figures 2, 3 and 8*), where the overall genotypic diversity is similar to the global diversity of *P. fluorescens*

(*Garrido-Sanz et al., 2016*). Pan-genome analysis and metagenomics represent increasingly powerful routes to understanding the genetic determinants of biological activity in plant-associated microbes (*Beskrovnaya et al., 2020*; *Biessy et al., 2019*; *Carrión et al., 2019*; *Melnyk et al., 2019*; *Mullins et al., 2019*; *Tracanna et al., 2021*).

Multiple BGCs and accessory genome loci were identified that correlated with on-plate inhibition of *S. scabies* growth and development (*Figure 3*), including BGCs for CLPs and HCN. These loci also correlated with inhibition of *P. infestans* and *G. graminis*, and their contribution to suppression was validated genetically. This confirmed a role for both molecules in *S. scabies* inhibition (*Figures 5 and 6*), representing a new function for these *Pseudomonas* specialized metabolites. Co-culture assays and cryo-SEM imaging (*Figure 6*) showed that HCN and a tensin-like CLP produced by Ps619 arrest the formation of streptomycete aerial hyphae and subsequent sporulation, providing the pseudomonad with a competitive advantage at the microbial interface.

In planta experiments confirmed that Ps619 could suppress potato scab and that CLP production was a key determinant of this inhibitory effect (*Figure 7*). In contrast, HCN production was not a requirement for potato scab suppression by Ps619. It is possible that HCN is not produced in sufficient amounts during root colonization for *S. scabies* inhibition, or that it instead has an alternative natural role, such as metal chelation (*Rijavec and Lapanje, 2016*). The roles of the CLPs are reminiscent of the interaction between *B. subtilis* and *S. coelicolor*, where the CLP surfactin functions as a surfactant required for the formation of aerial structures in *B. subtilis* and arrests aerial development in *S. coelicolor* (*Straight et al., 2006*). Collectively, these results are surprising given that streptomycetes themselves use surfactants to assist in the erection of aerial mycelia (*Kodani et al., 2004*; *Willey et al., 2006*) and points to secondary antagonistic roles for these molecules beyond the reduction of surface tension. This is strongly supported by the inhibitory effect of purified viscosin I towards *S. scabies* (*Figure 5C*).

HCN and CLPs have also been associated with insect (*Flury et al., 2017*) and nematode (*Siddiqui et al., 2006*) killing, as well as the suppression of pathogenic fungi (*Michelsen et al., 2015*; *Fukuda et al., 2021*). This indicates that a subset of pseudomonads are generalist suppressors of pathogens (and presumably also nonpathogenic organisms) due to the production of these broad range antimicrobials. Genetic analysis indicates that these strains are more likely to produce multiple suppressive metabolites and proteins. Evidence for this is provided by the inhibition of both *P. infestans* and *G. graminis* by Ps619 Δ*hcn*Δ*ten* (*Appendix 2—figure 1*). A study of the inhibitory properties of bacteria associated with the *Arabidopsis* leaf microbiome showed that a large proportion of the total inhibitory activity was due to *Pseudomonadales* bacteria and that a subset of individual strains were active against a wide array of bacteria (*Helfrich et al., 2018*).

Unexpectedly, irrigation led to a decrease in the proportion of suppressive pseudomonads on potato roots (*Figure 8A*) even though irrigation is one of the most effective ways to suppress potato scab. One possible reason for this discrepancy is that irrigation enables nonsuppressive *Pseudomonas* spp. with low motility to be transported to plant roots more effectively. Recruitment of nonsuppressive pseudomonads to the rhizosphere may benefit the plant in other ways, such as immune system priming (*Bakker et al., 2007*; *Teixeira et al., 2021*) or modulation of auxin biosynthesis. For example, *Cheng et al., 2017* showed that auxin biosynthesis was linked to plant growth promotion and induced systemic resistance by *P. fluorescens* SS101. An alternative hypothesis is that changes in the overall relative abundance of soil *Pseudomonas* over *Streptomyces* resulting from irrigation may override the observed shift towards less-suppressive *Pseudomonas* genotypes. In support of this, drought-induced enrichment for commensal *Streptomyces* and depletion of *Proteobacteria* in sorghum and rice plants have been shown to be reversed by irrigation (*Santos-Medellín et al., 2021*; *Xu et al., 2018*). In this model, irrigation may reduce the relative fitness of *S. scabies* versus *Pseudomonas* spp., while the microbiome of irrigated roots simultaneously becomes less optimal for disease suppression.

Our data show that Ps619 is highly effective at inhibiting potato scab, yet Ps619-like strains are naturally less abundant in irrigated conditions. Therefore, possible future efforts to control potato scab could combine irrigation with pretreatment with effective biocontrol strains, like Ps619, to ensure tubers are colonized by a significant proportion of biocontrol strains. Such a strategy could reduce the quantity of water required for effective scab suppression. While our study was focused on fluorescent pseudomonads, interactions between these bacteria and the wider microbiome (*Figure 1*) may also have a key role in potato scab suppression.

Moving forward, systematic analyses of individual organisms within microbiomes will continue to help answer questions relating to microbial communities and host interactions that are difficult to address using global 'omics approaches alone. For example, the role of many bacterial specialized metabolites in nature is poorly understood, especially for prolific producers such as the pseudomonads and the streptomycetes (*van der Meij et al., 2017*). Future studies could examine whether the host selects for bacterial populations enriched in specific BGCs and whether environmental stimuli modulate the abundance of these BGCs. Synthetic microbial communities based on well-characterized natural communities could then be used to test hypotheses on the role of specialized metabolites in shaping the community or modulating the health of the host organism.

# Materials and methods

## Key resources table

| Reagent type (species) or resource | Designation | Source or reference | Identifiers | Additional information |
|---|---|---|---|---|
| Strain, strain background (*Pseudomonas* spp.) | Ps | This paper | Ps616-Ps734 | 120 environmental *Pseudomonas* strains collected from RG Abrey Farms in February 2015 |
| Strain, strain background (*Pseudomonas* spp.) | Ps | This paper | Ps831-Ps950 | 120 environmental *Pseudomonas* strains collected from RG Abrey Farms in May 2015 |
| Strain, strain background (*Pseudomonas* spp.) | Ps | This paper | IR1-1-NS6-8 | 192 environmental *Pseudomonas* strains collected from RG Abrey Farms in June 2017 |
| Strain, strain background (*Pseudomonas fluorescens* SBW25) | WT | https://doi.org/10.1046/j.1365-2958.1996.391926.x (*Rainey and Bailey, 1996*) | SBW25 | Wild-type strain; viscosin producer |
| Strain, strain background (*Pseudomonas* sp.) | LMG 2338 | Belgian Coordinated Collections of Microorganisms (BCCM) | LMG 2338 NCPPB 387 | Wild-type strain; WLIP producer (*Mortishire-Smith et al., 1991*) |
| Strain, strain background (*Escherichia coli*) | DH5α | Thermo Fisher Scientific | 18265017 | Competent cells for cloning |
| Strain, strain background (*Streptomyces scabies*) | 87-22 | https://doi.org/10.1094/Phyto-85-537 | 87-22 | Causative agent of potato scab |
| Strain, strain background (*Phytophthora infestans*) | 6-A1 | The Sainsbury Laboratory, UK | #2006-3920A (6-A1) | Causative agent of late blight |
| Strain, strain background (*Gaeumannomyces graminis* var. *tritici*) | *Ggt* NZ.66.12 | https://doi.org/10.1111/1462-2920.13038 | NZ.66.12 | Causative agent of take-all decline in wheat |
| Genetic reagent (*Pseudomonas* sp. Ps682) | Δvisc | This paper | PS682_04206 | In-frame deletion of PS682_04206 (VVN17163.1) in the viscosin-like BGC using allelic exchange |
| Genetic reagent (*Pseudomonas* sp. Ps682) | ::lux | This paper | luxCDABE | Introduction of the *Aliivibrio fischeri* luxCDABE cassette into the neutral att::Tn7 site of the Ps682 chromosome using the Tn7-based expression system (K.-H. *Choi et al., 2005*) |
| Genetic reagent (*Pseudomonas* sp. Ps682) | Δvisc::lux | This paper | PS682_04206 luxCDABE | Introduction of the *A. fischeri* luxCDABE cassette into the neutral att::Tn7 site of the Ps682 Δvisc chromosome using the Tn7-based expression system (K.-H. *Choi et al., 2005*) |
| Genetic reagent (*Pseudomonas* sp. Ps619) | Δten | This paper | PS619_02963 | In-frame deletion of PS619_02963 (VVM93793.1) in the tensin-like BGC using allelic exchange |
| Genetic reagent (*Pseudomonas* sp. Ps619) | Δhcn | This paper | PS619_05844 (hcnB) PS619_05845 (hcnC) | In-frame deletion of PS619_05844 (VVN46770.1) and PS619_05845 (VVN46780.1) in the HCN BGC using allelic exchange |
| Genetic reagent (*Pseudomonas* sp. Ps619) | ΔtenΔhcn | This paper | PS619_02963 PS619_05844 (hcnB) PS619_05845 (hcnC) | In-frame deletions of PS619_02963 in the tensin-like BGC and PS619_05844 to PS619_05845 in the HCN BGC using allelic exchange |

*Continued on next page*

*Continued*

| Reagent type (species) or resource | Designation | Source or reference | Identifiers | Additional information |
|---|---|---|---|---|
| Biological sample (*Solanum tuberosum*) | Potato seeds cv. Maris Piper | VCS Potatoes Ltd. | | Seed potatoes used for potato scab infection assays |
| Recombinant DNA reagent | pTS1 | https://doi.org/101038/ncomms15935 | | pME3087 derivative containing a *sacB* counter-selection marker |
| Recombinant DNA reagent | pTS1-Δviscosin | This paper | PS682_04206 | Plasmid for PS682_04206 deletion in viscosin-like BGC of Ps682 |
| Recombinant DNA reagent | pTS1-Δtensin | This paper | PS619_02963 | Plasmid for PS619_02963 deletion in tensin-like BGC of Ps619 |
| Recombinant DNA reagent | pTS1-Δ619HCN | This paper | PS619_05844 PS619_05845 | Plasmid for deletion of PS619_05844 to PS619_05845 in HCN BGC of Ps619 |
| Recombinant DNA reagent | pTNS2 | https://doi.org/10.1038/nmeth765 | | Tn7 transposase expression plasmid |
| Recombinant DNA reagent | pUC18-mini-Tn7T-Gm-lux | https://doi.org/10.1038/nmeth765 | *luxCDABE* | mini-Tn7 *luxCDABE* transcriptional fusion vector |
| Sequence-based reagent | PCR primers | This paper | | Primers used in this study are listed in **Supplementary file 2B** |
| Sequence-based reagent | F515/R806 | https://doi.org/10.1073/pnas.1000080107 | F515 R806 | Primer pair for amplicon sequencing of the v4 region of 16S rRNA |
| Commercial assay or kit | FastDNA SPIN Kit for Soil | MP Biomedicals | 116560200 | DNA extraction from soil samples |
| Commercial assay or kit | GenElute Bacterial Genomic DNA Kit | Sigma-Aldrich | NA2110 | Genomic DNA extraction from isolated bacteria |
| Chemical compound, drug | Copper(II) ethyl acetoacetate | Sigma-Aldrich | 731714 | Reagent for Feigl–Anger assay of HCN production |
| Chemical compound, drug | 4,4'-Methylenebis N,N-dimethylaniline | Sigma-Aldrich | M44451 | Reagent for Feigl–Anger assay of HCN production |
| Software, algorithm | MaSuRCA | https://doi.org/10.1093/bioinformatics/btt476 | RRID:SCR_010691 Version 3.2.6 | Genome assembly; https://github.com/alekseyzimin/masurca |
| Software, algorithm | SPAdes | https://doi.org/10.1089/cmb.2012.0021 | RRID:SCR_000131 Version 3.6.2 | Genome assembly; https://github.com/ablab/spades |
| Software, algorithm | Prokka | https://doi.org/10.1093/bioinformatics/btu153 | RRID:SCR_014732 Version 1.14.0 | Genome annotation; https://github.com/tseemann/prokka |
| Software, algorithm | CheckM | https://doi.org/10.1101/gr.186072.114 | RRID:SCR_016646 Version 1.1.3 | Quality control assessment of bacterial genomes; https://ecogenomics.github.io/CheckM/ |
| Software, algorithm | antiSMASH | https://doi.org/10.1093/nar/gkz310 | Version 5.0 | Biosynthetic gene cluster detection and analysis; https://antismash.secondarymetabolites.org |
| Software, algorithm | MultiGeneBlast | https://doi.org/10.1093/molbev/mst025 | | BLAST searches for gene clusters; http://multigeneblast.sourceforge.net/ |
| Software, algorithm | MUSCLE | https://doi.org/10.1093/nar/gkh340 | Version 3.8.31 | Sequence alignment; https://www.drive5.com/muscle/ |
| Software, algorithm | RAxML | https://doi.org/10.1093/bioinformatics/btu033 | RRID:SCR_006086 Version 8.2.12 | Phylogenetic analysis; https://github.com/stamatak/standard-RAxML |
| Software, algorithm | Interactive Tree of Life (iTOL) | https://doi.org/10.1093/nar/gkab301 | Version 5 | Visualization of phylogenetic trees; https://itol.embl.de/ |

*Continued on next page*

*Continued*

| Reagent type (species) or resource | Designation | Source or reference | Identifiers | Additional information |
|---|---|---|---|---|
| Software, algorithm | Global Natural Product Social Molecular Networking (GNPS) | https://doi.org/10.1038/nbt.3597 | | Networking of mass spectrometry data; https://gnps.ucsd.edu |
| Software, algorithm | Cytoscape | https://doi.org/10.1101/gr.1239303 | Version 3.8.2 | Visualization of networks; https://cytoscape.org/ |
| Software, algorithm | TopSpin | Bruker | Version 3.5 | NMR data analysis |
| Software, algorithm | Mnova 14.0 | Mestrelab Research | Version 14.0 | NMR data analysis |
| Software, algorithm | R | The R foundation | Version 3.5.1 | Data analysis; https://www.r-project.org/ |

## Strains and growth conditions

All strains used in this study are listed in *Supplementary file 2A*. Unless figure otherwise stated, chemicals were purchased from Sigma-Aldrich, enzymes from New England Biolabs, and molecular biology kits from GE Healthcare and Promega. All *P. fluorescens* strains were grown at 28°C in L medium (Luria base broth, Formedium) and *Escherichia coli* at 37°C in lysogeny broth (LB) (*Miller, 1972*). 1.3% agar was added for solid media. Gentamicin was used at 25 µg/mL, carbenicillin at 100 µg/mL, and tetracycline (Tet) at 12.5 µg/mL. *S. scabies* spore suspensions were prepared using established procedures (*Kieser et al., 2000*).

## Soil sample collection

Soil samples were collected from potato fields at RG Abrey Farms (East Wretham, Norfolk, UK, 52.4644° N, 0.8299° E). The first sampling was conducted in 2015 from two adjacent plots in a single field. Soil samples were taken on 22 January 2015, immediately prior to planting. One plot was then covered loosely in polythene to protect it from irrigation. The same field sites were sampled again in May at the point of maximum scab impact, once potato tubers had begun to form. In this case, soil samples were taken from the base of the plants, near the root system. For each sampling event, a total of 12 samples were taken from three parallel potato beds at regularly spaced intervals approximately 1 m apart. For the second experiment, 12 irrigated and 12 nonirrigated potato plants were uprooted from field sites in June 2017 and returned to the laboratory in large pots. Bulk soil samples were taken from these pots alongside an equivalent number of rhizosphere-associated samples, which were defined as isolated root systems gently shaken to remove bulk soil before processing as below. Samples were collected in sterile 50 mL tubes and stored at 4°C.

## Isolation of soil *Pseudomonas*

Sample processing was conducted at 4°C throughout. 10 mL of sterile phosphate-buffered saline (PBS, per liter: 8 g NaCl, 0.2 g KCl, 1.44 g $Na_2HPO_4$, 0.24 g $KH_2PO_4$, pH 7.4) were added to 50 mL tubes containing 20 g of soil or root material, and vortexed vigorously for 10 min. Samples were then filtered through a sterile muslin filter to remove larger debris. The resulting suspension of soil and organic matter was centrifuged at 1000 rpm for 30 s to pellet remaining soil particles, before serial dilution in PBS and plating on *Pseudomonas* selective agar. The selection media comprised *Pseudomonas* agar base (Oxoid, UK) supplemented with CFC (cetrimide/fucidin/cephalosporin) *Pseudomonas* selective supplement (Oxoid, UK). Plates were incubated at 28°C until colonies arose, then isolated single colonies were patched on fresh CFC agar and incubated overnight at 28°C before streaking to single colonies on King's B (KB) agar plates (*King et al., 1954*). Six isolates were selected at random per soil sample and subjected to phenotypic/genomic analysis.

## Amplicon sequencing

Genomic DNA was isolated from 3 g of pooled soil samples using the FastDNA SPIN Kit for soil (MP Biomedicals, UK) following the manufacturer's instructions. Genomic DNA concentration and purity was determined by NanoDrop spectrophotometry as above. Microbial 16S rRNA genes were amplified from soil DNA samples with barcoded universal prokaryotic primers (F515/R806) targeting the V4 region (*Caporaso et al., 2011*), and then subjected to Illumina MiSeq sequencing (600-cycle, 2 × 300 bp) at the DNA Sequencing Facility, Department of Biochemistry, University of Cambridge

(UK). The data were analyzed using the MiSeq Reporter Metagenomics Workflow (Illumina, UK) to acquire read counts for all taxonomic ranks from phylum to genus. MiSeq data were visualized and analyzed using Degust 3.1.0 (http://degust.erc.monash.edu/) and Pheatmap (https://CRAN.R-project.org/package=pheatmap) in R 3.5.1.

## Phenotypic assays

All phenotyping assays were conducted at least twice independently, and where disagreements were recorded in the ordinal data, additional repeats were conducted until a firm consensus was reached.

### Swarming motility

0.5% KB agar plates were poured and allowed to set and dry for 1 hr in a sterile flow cabinet. Plates were then inoculated with 2 µL spots of overnight cultures and incubated overnight at room temperature. The motility of each isolate was tested in triplicate and scored from 0 (no motility) to 3 (high motility).

### Secreted protease activity

5 µL of overnight cultures were spotted onto KB plates containing 1.0% skimmed milk powder. Plates were incubated at 28°C and photographed after 24 hr, with individual isolates scored as protease positive (score = 2) or negative (score = 0).

### HCN production

An adaptation of the method described in *Castric and Castric, 1983* was used. *Pseudomonas* isolates were inoculated into 150 µL of liquid KB medium in individual wells of a flat-bottomed 96-well plate. The plates were then overlaid with Feigl–Anger reagent paper (*Feigl and Anger, 1966*), prepared as follows. Whatman 3MM chromatography paper was soaked in Feigl–Anger detection reagent (5 mg/mL copper(II) ethyl acetoacetate and 5 mg/mL 4,4'-methylenebis(N,N-dimethylaniline) dissolved in chloroform). After complete solvent evaporation, the paper was placed under the plate lid and strains were grown at 28°C overnight with gentle shaking. The intensity of blue staining on the paper was then scored from 0 (no color) to 3 (high blue intensity). The same method was applied to isolates growing on agar medium.

### *S. scabies* inhibition

Two parallel lines of *S. scabies* 87-22 spores were streaked onto SFM plates (*Kieser et al., 2000*) using a sterile toothpick. These lines were then cross-streaked with overnight cultures of *Pseudomonas* isolates. Plates were incubated at 30°C, and the relative performance of each species was assessed daily for 5 days.

### *P. infestans* inhibition

Assays were conducted with *P. infestans* #2006-3920A (6-A1) (The Sainsbury Laboratory, UK). This was maintained on rye agar medium supplemented with 2% sucrose (C-RSA) (*Caten and Jinks, 1968*) at 21°C. C-RSA was filtered through muslin fabric to enable clearer observation of oomycete growth. Three 10 µL drops of overnight cultures per *Pseudomonas* isolate strain were placed equidistantly 15 mm from the edge of C-RSA plates. A 3 mm plug from the leading edge of a *P. infestans* culture was then placed in the center of each plate and incubated for a further 7 days at 21°C before scoring and imaging.

### Take-all inhibition

Assays were conducted with *G. graminis* var. *tritici* strain NZ.66.12 (*Ggt*) (*Mauchline et al., 2015*). Three 10 µL drops of overnight *Pseudomonas* cultures per strain were placed equidistantly 15 mm from the edge of potato dextrose agar (PDA) plates and incubated for 24 hr at 28°C. *Ggt* NZ.66.12 was cultured on PDA agar for 5 days at room temperature. A 3 mm plug from the leading edge of the NZ.66.12 culture was then placed in the center of each plate and incubated for a further 5 days at 22°C before the extent of *Ggt* inhibition was assessed.

## DNA extraction and Illumina genome sequencing

Single colonies of each isolate to be sequenced were picked from L agar plates and grown overnight in L medium. DNA was then extracted from 2 mL of cell culture using a GenElute Bacterial Genomic DNA Kit (Sigma-Aldrich, USA). DNA samples were subjected to an initial quality check using a Nano-Drop spectrophotometer (Thermo Scientific, Wilmington, DE) before submission for Nextera library preparation and paired-end read sequencing on the Illumina MiSeq platform (600-cycle, 2 × 300 bp) at the DNA Sequencing Facility, Department of Biochemistry, University of Cambridge (UK). Reads from 35 pseudomonads collected in February 2015 were assembled into genomes using MaSuRCA v3.2.6 (**Zimin et al., 2013**) with the following settings:

GRAPH_KMER_SIZE = auto; USE_LINKING_MATES = 1; LIMIT_JUMP_COVERAGE = 60; CA_PARAMETERS = ovlMerSize = 30 cgwErrorRate = 0.25 ovlMemory = 4 GB; NUM_THREADS = 16; JF_SIZE = 100000000; DO_HOMOPOLYMER_TRIM = 0.

Reads from 32 samples collected in May 2015 were assembled into genomes using SPAdes v3.6.2 (**Bankevich et al., 2012**) with k-mer flag set to -k 2133557799127. All assembly tasks were conducted using 16 CPUs on a 256 GB compute node within the Norwich Bioscience Institutes (NBI) High Performance Computing cluster. An additional strain from May 2015 (Ps925) was sequenced and assembled by MicrobesNG (http://www.microbesng.uk), which is supported by the BBSRC (grant number BB/L024209/1). The 69 assembled genome sequences were annotated using Prokka (**Seemann, 2014**), which implements Prodigal (**Hyatt et al., 2010**) as an open-reading frame calling tool. Assembly qualities were assessed using CheckM (**Parks et al., 2015**). Genome assemblies are available at the European Nucleotide Archive (http://www.ebi.ac.uk/ena/) with the project accession PRJEB34261.

## Phylogenetic and bioinformatic analysis

The *gyrB* housekeeping gene sequence was identified in each newly sequenced genome by BLAST comparison with the sequence of *gyrB* from *P. fluorescens* SBW25. The full-length *gyrB* sequences from these strains and several reference strains were aligned using MUSCLE 3.8.31 (**Edgar, 2004**) with default settings, then a maximum likelihood tree was calculated using RAxML 8.2.12 (**Stamatakis, 2014**) on the CIPRES portal (**Miller et al., 2015**) with the following parameters: raxmlHPC-HYBRID-AVX -T 4f a -N autoMRE -n result -s infile.txt -c 25 m GTRCAT -p 12345k -x 12345. Genomes were subjected to bioinformatic analysis as described in Appendix 1. Phylogenetic trees and presence/absence data for accessory genes were visualized using Interactive Tree of Life (iTOL) (**Letunic and Bork, 2016**), with *Pseudomonas aeruginosa* PAO1 *gyrB* as the outgroup.

## Molecular biology procedures

Cloning was carried out in accordance with standard molecular biology techniques. *P. fluorescens* deletion mutants were constructed by allelic exchange as described previously (**Campilongo et al., 2017**). Up- and downstream flanking regions (approximately 500 bp) to the target genes were amplified using primers listed in **Supplementary file 2B**. PCR products in each case were ligated into pTS1 (**Scott et al., 2017**) between XhoI and BamHI. The resulting deletion vectors were transformed into the target strains by electroporation, and single crossovers selected on L + Tet and re-streaked to isolate single colonies. 100 mL cultures in L medium from single crossovers were grown overnight at 28°C, then plated onto L + 10% sucrose plates to counter-select for double crossovers. Individual colonies from these plates were then patched onto L plates ± Tet, with Tet-sensitive colonies tested for gene deletion by colony PCR using primers external to the deleted gene in each case (**Supplementary file 2B**).

Luminescent-tagged strains were produced by introduction of the *Aliivibrio fischeri luxCDABE* cassette into the neutral *att::Tn7* site in *Pseudomonas* chromosomes using the Tn7-based expression system described in **Choi et al., 2005**. Strains were electroporated with plasmids pUC18-mini*Tn7*T-Gm-*lux* and the helper pTNS2, and transformant colonies were grown on solid L medium+ gentamicin for 2–3 days at 28°C. Integration of the *lux* cassette into *Pseudomonas* genomes was confirmed by PCR and with a luminometer. Luminescent cells were then tracked using the NightOWL visualization system (Berthold Technologies, Germany). All plasmids used in this study are reported in **Supplementary file 2C**.

## LC-MS detection of lipopeptides

*Pseudomonas* isolates were grown overnight in L medium (10 mL) for 16 hr at 28°C. 100 µL of each culture was used to inoculate 40 mm diameter KB agar plates. Plates were incubated for 24 hr at 28°C, before the agar from each plate was decanted into a sterile 50 mL tube and extracted with 10 mL 50% EtOH with occasional vortexing for 3 hr. 2 mL was taken from each sample and centrifuged in 2 mL tubes for 5 min at 16,000 × *g*. The supernatant was collected and stored at –80°C. Samples were diluted with an equal volume of water, then subjected to LC-MS analysis using a Shimadzu Nexera X2 UHPLC coupled to a Shimadzu ion-trap time-of-flight (IT-TOF) mass spectrometer. Samples (5 µL) were injected onto a Phenomenex Kinetex 2.6 µm XB-C18 column (50 × 2.1 mm, 100 Å), eluting with a linear gradient of 5–95% acetonitrile in water +0.1% formic acid over 6 min with a flow rate of 0.6 mL/min at 40°C. To compare the retention times of viscosin I (Ps682), WLIP (*Pseudomonas* sp. LMG 2338), and viscosin (*P. fluorescens* SBW25), extracts were prepared from their producing organisms as described above. The same chromatography conditions as above were used, but with a linear gradient of 5–100% acetonitrile in water + 0.1% formic acid over 15 min.

Positive mode mass spectrometry data were collected between *m/z* 300 and 2000 with an ion accumulation time of 10 ms featuring an automatic sensitivity control of 70% of the base peak. The curved desolvation line temperature was 300°C, and the heat block temperature was 250°C. MS/MS data were collected in a data-dependent manner using collision-induced dissociation energy of 50% and a precursor ion width of 3 Da. The instrument was calibrated using sodium trifluoroacetate cluster ions prior to every run.

A molecular network was created using the online workflow at the Global Natural Product Social Molecular Networking (GNPS) site (https://gnps.ucsd.edu/; *Aron et al., 2020*). The data were filtered by removing all MS/MS peaks within ±17 Da of the precursor *m/z*. The data were then clustered with MS-Cluster with a parent mass tolerance of 1 Da and an MS/MS fragment ion tolerance of 0.5 Da to create consensus spectra. Consensus spectra that contained less than two spectra were discarded. A network was then created where edges were filtered to have a cosine score above 0.6 and more than four matched peaks. Further edges between two nodes were kept in the network if each of the nodes appeared in each other's respective top 10 most similar nodes. The spectra in the network were then searched against GNPS spectral libraries. The library spectra were filtered in the same manner as the input data. All matches kept between network spectra and library spectra were required to have a score above 0.7 and at least four matched peaks. Networks were visualized using Cytoscape v3.8.2 (*Shannon et al., 2003*), and the data were manually filtered to remove duplicate nodes (same *m/z* and retention time). The data are available as MassIVE dataset MSV000084283 at https://massive.ucsd. edu, and the GNPS analysis is available at https://gnps.ucsd.edu/ProteoSAFe/status.jsp?task=51ac 5fe596424cf88cfc17898985cac2.

High-resolution mass spectra were acquired on a Synapt G2-Si mass spectrometer equipped with an Acquity UPLC (Waters). Aliquots of the samples were injected onto an Acquity UPLC BEH C18 column, 1.7 µm, 1 × 100 mm (Waters) and eluted with a gradient of acetonitrile/0.1% formic acid (B) in water/0.1% formic acid (A) with a flow rate of 0.08 mL/min at 45°C. The concentration of B was kept at 1% for 1 min followed by a gradient up to 40% B in 9 min, ramping to 99% B in 1 min, kept at 99% B for 2 min and re-equilibrated at 1% B for 4 min. MS data were collected in positive mode with the following parameters: resolution mode, positive ion mode, scan time 0.5 s, mass range *m/z* 50–1200 calibrated with sodium formate, capillary voltage = 2.5 kV; cone voltage = 40 V; source temperature = 125°C; desolvation temperature = 300°C. Leu-enkephalin peptide was used to generate a lock-mass calibration with 556.2766, measured every 30 s during the run. For MS/MS fragmentation, a data-directed analysis (DDA) method was used with the following parameters: precursor selected from the four most intense ions; MS2 threshold: 5000; scan time 0.5 s; no dynamic exclusion. In positive mode, collision energy (CE) was ramped between 10–30 at low mass (*m/z* 50) and 15–60 at high mass (*m/z* 1200).

## Purification and characterization of viscosin I

Pre-cultures of Ps682 were grown in 10 mL LB medium for 16 hr and 600 µL aliquots were used to inoculate 140 mm diameter KB agar plates. Fifteen plates were inoculated and incubated for 24 hr. The agar was decanted and extracted with 500 mL ethyl acetate for 2 hr with occasional mixing. The organic fraction was filtered off, washed with 3 × 200 mL water, dried over MgSO₄, and then solvent

was removed *in vacuo*. The resulting material was dissolved in MeOH and applied to a 12 g C18 flash chromatography column (Biotage), pre-equilibrated in 70% MeOH. Separation proceeded by a gradient of 70–100% MeOH over 10-column volumes. Each fraction was subject to LC-MS analysis using a Shimadzu Nexera X2 UHPLC coupled to a Shimadzu IT-TOF mass spectrometer, as described above.

Solvent was removed from viscosin I-containing fractions using a rotary evaporator and then a Genevac (SP Scientific). Fractions were then dissolved in MeOH to 1 mg/mL and further purified using a Thermo Dionex Ultimate 3000 HPLC system. 200 µL aliquots were injected onto a Phenomenex C18 Luna column (5 µm, 250 mm × 10 mm) and eluted with a linear gradient of 5–95% acetonitrile/H$_2$O over 30 min, with a flow rate of 4 mL/min and UV absorption data collected at 210 nm. LC-MS analysis (as described above) was used to identify pure fractions, which were combined and dried *in vacuo* to yield 1.4 mg viscosin I as a white powder.

Viscosin I (1.4 mg) was dissolved in *N,N*-dimethylformamide-d$_7$ (DMF-d$_7$) and NMR spectra were acquired on a Bruker Avance Neo 600 MHz spectrometer equipped with a TCI cryoprobe. The experiments were carried out at 298 K with the residual DMF solvent used as an internal standard ($\delta_H/\delta_C$ 2.75/29.76). The residual solvent signal from H$_2$O was suppressed through a presaturation sequence in 1D $^1$H. Resonances were assigned through 1D $^1$H and DEPT135 experiments, and 2D COSY, HSQCed, HMBC, TOCSY, and HSQC-TOCSY experiments. Spectra were analyzed using Bruker TopSpin 3.5 and Mestrelab Research Mnova 14.0 software. NMR data are reported in *Figure 5—figure supplements 3–13* and *Supplementary file 2D*.

## Disk diffusion assays

A *S. scabies* 87-22 spore suspension was diluted 1:100 in sterile MQ water and 60 µL aliquots were applied to instant potato medium (20 g/L Smash Instant Mash, 20 g/L agar) on 100 mm square plates. The spore solution was evenly distributed using a sterile cotton bud and the plate was dried for 30 min. Viscosin I was diluted in MeOH to produce a range of concentrations from 20 to 100 µg/mL. Each concentration was applied to a 6 mm filter paper disk, in 5 × 20 µL applications at 10 min intervals, and then dried for 30 min. The disks were then applied to the surface of the agar plate, which was incubated at 30°C and imaged daily.

## Scanning electron microscopy

Small pieces of the *Pseudomonas-Streptomyces* co-culture samples were excised from the surface of agar plates and mounted on an aluminum stub using Tissue Tek (BDH Laboratory Supplies, Poole, England). The stub was then immediately plunged into liquid nitrogen slush at approximately –210°C to cryo-preserve the material. The sample was transferred onto the cryostage of an ALTO 2500 cryo-transfer system (Gatan, Oxford, England) attached to an FEI Nova NanoSEM 450 (FEI, Eindhoven, The Netherlands). Sublimation of surface frost was performed at –95°C for 3 min before sputter coating the sample with platinum for 3 min at 10 mA, at colder than –110°C. After sputter coating, the sample was moved onto the cryo-stage in the main chamber of the microscope, held at –125°C. The sample was imaged at 3 kV and digital TIFF files were stored.

## Potato scab biocontrol assays

In planta assays were performed as described previously (*Lin et al., 2018*; *Sarwar et al., 2018*) with some modifications. Briefly, 2 L of GYM (4 g/L glucose, 4 g/L yeast extract, 10 g/L malt extract, 2 g/L CaCO$_3$, pH 7.2) was inoculated with a *S. scabies* 87-22 starter culture from a spore suspension and incubated for 48 hr at 30°C, 250 rpm. Each culture was then centrifuged at 16,994 × *g* for 15 min and washed twice with PBS (2 L). *Pseudomonas* strains Ps619, Ps682, and associated mutants were grown overnight at 28°C, 250 rpm in 50 mL L medium (Luria base broth, Formedium), centrifuged at 1520 × *g* for 15 min, washed twice with PBS (20 mL), and adjusted to OD$_{600}$ = 0.2 for the final inoculation.

50 mL of autoclaved vermiculite and 50 mL of bacterial culture were mixed to constitute the final inoculum. 5 L pots were filled with steam-sterilized substrate (John Innes Cereal Mix) and inoculum was applied into the pots and mixed with the soil. Different combinations of bacterial inocula were made accordingly following the same method. Potato seeds cv. Maris Piper obtained from VCS Potatoes Ltd (Suffolk, UK) were surfaced-disinfected by immersion in 1% sodium hypochlorite for 15 min. Tubers were then rinsed with water, air-dried, and placed in pots. Pots were watered until saturation

according to their growth stage. Potato plants were grown in a glasshouse with a light cycle of 16 h/8 h at 18–20°C. Two independent experiments were run between 17 July and 6 November 2020 and 31 July and 20 November 2020, and 3–4 plants were used per treatment. Tubers were collected, washed, weighed, and scored accordingly to a 1–6 scale as described in *Andrade et al., 2019*. Potato plants were dried for 4 days at 30°C and aerial parts and tuber weights were both recorded, as well as tuber number. Treatment differences were carried out based on the disease index (DI) of each plant (n = 4 for each treatment). The DI was calculated as the mean of all the scored tubers per plant, and p-values were calculated using Dunnett's multiple-comparison test.

## Acknowledgements

Financial support was provided by a Royal Society University Research Fellowship to AT, Biotechnology and Biological Sciences Research Council (BBSRC) BIO, MET, PH, and MfN Institute Strategic Programme grants to the John Innes Centre (JIC), a JIC Institute Development Grant and NPRONET Proof of Concept grant. JF was supported by a BBSRC DTP studentship, and AP-M was supported by a BBSRC iCASE PhD studentship, both awarded to the Norwich Research Park. We thank Prof. Jonathan Jones (The Sainsbury Laboratory) for providing *P. infestans* isolates and Dr Tim Mauchline (Rothamsted Research) for providing *Ggt* NZ.66.12. We also thank the JIC Bioimaging, Metabolomics, and NMR facilities for their contribution to this publication, Dr Carlo de Oliveira Martins (JIC) for assistance with mass spectrometry, and Dr Natalia Miguel-Vior (JIC) for assistance with potato scab biocontrol assays.

## Additional information

### Competing interests

Graham Tomalin: affiliated with VCS Potatoes and has no financial interests to declare. The other authors declare that no competing interests exist.

### Funding

| Funder | Grant reference number | Author |
| --- | --- | --- |
| Biotechnology and Biological Sciences Research Council | BB/J004596/1 | Andrew W Truman |
| Biotechnology and Biological Sciences Research Council | BBS/E/J/000PR9790 | Andrew W Truman |
| Biotechnology and Biological Sciences Research Council | BB/J004553/1 | Jacob G Malone |
| Biotechnology and Biological Sciences Research Council | BBS/E/J/000PR9797 | Jacob G Malone |
| Biotechnology and Biological Sciences Research Council | BB/M011216/1 | Alba Pacheco-Moreno Jonathan J Ford |
| NPRONET | POC021 | Graham Tomalin Jacob G Malone Andrew W Truman |
| Royal Society | URF\R\180007 | Andrew W Truman |

The funders had no role in study design, data collection and interpretation, or the decision to submit the work for publication.

## Author contributions
Alba Pacheco-Moreno, Formal analysis, Investigation, Methodology, Visualization, Writing – review and editing; Francesca L Stefanato, Formal analysis, Investigation, Methodology, Validation, Visualization, Writing – review and editing; Jonathan J Ford, Investigation, Visualization, Writing – review and editing; Christine Trippel, Simon Uszkoreit, Data curation, Investigation, Methodology; Laura Ferrafiat, Lucia Grenga, Ruth Dickens, Nathan Kelly, Liana Ambrosetti, Investigation; Alexander DH Kingdon, Formal analysis, Investigation; Sergey A Nepogodiev, Formal analysis, Supervision, Visualization; Kim C Findlay, Investigation, Visualization; Jitender Cheema, Formal analysis; Martin Trick, Govind Chandra, Data curation, Formal analysis; Graham Tomalin, Project administration, Writing – review and editing; Jacob G Malone, Conceptualization, Funding acquisition, Methodology, Project administration, Supervision, Visualization, Writing – original draft, Writing – review and editing; Andrew W Truman, Conceptualization, Formal analysis, Funding acquisition, Methodology, Project administration, Supervision, Visualization, Writing – original draft, Writing – review and editing

## Author ORCIDs
Francesca L Stefanato http://orcid.org/0000-0002-7961-6478
Jonathan J Ford http://orcid.org/0000-0001-9886-690X
Lucia Grenga http://orcid.org/0000-0001-5560-1717
Alexander DH Kingdon http://orcid.org/0000-0001-7074-6893
Sergey A Nepogodiev http://orcid.org/0000-0001-9796-4612
Kim C Findlay http://orcid.org/0000-0002-1556-0532
Govind Chandra http://orcid.org/0000-0002-7882-6676
Jacob G Malone http://orcid.org/0000-0003-1959-6820
Andrew W Truman http://orcid.org/0000-0001-5453-7485

## Decision letter and Author response
Decision letter https://doi.org/10.7554/eLife.71900.sa1
Author response https://doi.org/10.7554/eLife.71900.sa2

# Additional files

## Supplementary files
• Supplementary file 1. XLSX file containing genotypic and phenotypic data associated with *Figures 2, 3 and 8*.
• Supplementary file 2. DOCX file containing Supplementary files 2A–D.
• Transparent reporting form

## Data availability
Genome assemblies are available at the European Nucleotide Archive (http://www.ebi.ac.uk/ena/) with the project accession PRJEB34261. Mass spectrometry data are available as MassIVE dataset MSV000084283 at https://massive.ucsd.edu and the GNPS analysis is available here: https://gnps.ucsd.edu/ProteoSAFe/status.jsp?task=51ac5fe596424cf88cfc17898985cac2. All other data generated in this study are included in the manuscript and supporting files.

The following dataset was generated:

| Author(s) | Year | Dataset title | Dataset URL | Database and Identifier |
| --- | --- | --- | --- | --- |
| Pacheco-Moreno A, Stefanato FL, Ford JF, Trippel C, Uszkoreit S, Ferrafiat L, Grenga L, Dickens R, Kelley N, Kingdon ADH, Ambrosetti L, Nepogodiev SA, Findlay KC, Cheema J, Trick M, Chandra G, Tomalin G, Malone JG, Truman AW | 2019 | Pan-genome analysis identifies intersecting roles for Pseudomonas specialized metabolites in potato pathogen inhibition | https://www.ebi.ac. uk/ena/browser/view/ PRJEB34261 | European Nucleotide Archive, PRJEB34261 |

*Continued on next page*

*Continued*

| Author(s) | Year | Dataset title | Dataset URL | Database and Identifier |
|---|---|---|---|---|
| Pacheco-Moreno A, Stefanato FL, Ford JF, Trippel C, Uszkoreit S, Ferrafiat L, Grenga L, Dickens R, Kelley N, Kingdon ADH, Ambrosetti L, Nepogodiev SA, Findlay KC, Cheema J, Trick M, Chandra G, Tomalin G, Malone JG, Truman AW | 2021 | Pseudomonas strains isolated from a potato field in East Anglia | https://massive.ucsd.edu/ProteoSAFe/dataset.jsp?task=dc75ea1d4a8e438098569cf75a93bee1 | MassIVE, MSV000084283 |

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

# Appendix 1

## Identification and scoring of biosynthetic gene clusters and accessory genome loci

All genome sequences were subjected to BGC analysis using antiSMASH 5.0 (*Blin et al., 2019*). Any 'similar known cluster' and/or MIBiG BGC-ID annotations (*Kautsar et al., 2020*) from antiSMASH were assessed for their accuracy via a manual comparison with the putative matching BGC. This was particularly important for BGCs that are split across distinct genomic loci, such as pyoverdine and viscosin (*Gross and Loper, 2009*). When a characterized homologous BGC could not be identified, the BGCs were named and numbered based on their biosynthetic class (e.g., NRPS 1, NRPS 2, NRPS 3). To ensure that no BGCs had been missed by antiSMASH analysis, all genomes were searched against a library of known *Pseudomonas* BGCs using MultiGeneBlast (*Medema et al., 2013*) (settings: minimal sequence coverage of BLAST hits = 25%; minimal identity of BLAST hits = 25%; maximal distance between genes in locus = 20 kb). MultiGeneBlast was also used to validate the antiSMASH annotations. Brief details of each BGC are summarized below and were scored as 0 (no BGC), 1 (partial BGC), or 2 (full BGC). Scoring criteria vary between BGC type in relation to the requirements for a functional BGC in each case (details below). Where conserved domains were not evident in an antiSMASH output, their identity and specificity (for NRPS/PKS domains) were further assessed using NCBI CD-Search (*Marchler-Bauer et al., 2017*) and NRPSpredictor2 (*Röttig et al., 2011*). When a single protein defined a genotype (such as secreted proteases), BLAST analysis (*Camacho et al., 2009*) was carried out to identify homologues. Results are summarized in *Supplementary file 1*.

Key to domain nomenclature for PKSs and NRPSs: A = adenylation; T = thiolation; C = condensation; TE = thioesterase; AT = acyltransferase; KS = keto synthase; KR = ketoreductase; DH = dehydratase; ER = enoylreductase; MT = methyltransferase. / = denotes the boundary between PKS or NRPS proteins.

## NRPSs

Note that some NRPS BGCs are described in the siderophore section, such as pyoverdine.

Cyclic and linear lipopeptides: These were subjected to detailed analysis as reported elsewhere in Materials and methods.

Safracin: A well-characterized antitumor compound produced by an NRPS (*Velasco et al., 2005*). The full *ten* gene BGC is present in the Ps903 genome.

*mgo/pvf* operon: A conserved cluster that includes a single-module NRPS protein with A-T-reductase domain organization and predicted leucine specificity. Initially identified as being involved in mangotoxin biosynthesis, the *mgo* operon is distinct from the *mbo* operon, which is actually responsible for mangotoxin biosynthesis (*Carrión et al., 2013*). The *mgo* operon is homologous to the *Pseudomonas entomophila* virulence factor (*pvf*) gene cluster, which is a regulator of virulence factors (*Kretsch et al., 2021*; *Vallet-Gely et al., 2010*). This indicates that the product of this BGC functions as a regulator of other BGCs in *Pseudomonas* species.

NRPS 1: Trimodular NRPS (A-T-C-A-T-C-A-T-TE) with Thr-Thr-? predicted specificity.

NRPS 2: Three co-transcribed proteins featuring six NRPS modules plus a single PKS lacking an AT (A-T-C-A-T-C-A-T / C-A-T-TE / A-T-C-A-T-KS-KR-T-TE; predicted specificity = Arg-Pro-Cys /?-TE / Val-Pro-?-TE). Two TE domains indicate that this could produce two metabolites. Wider cluster includes an MbtH-like protein and a dioxygenase.

NRPS 3: Single protein containing a single NRPS module (A-T-TE) with no known amino acid specificity. Adjacent to a glutathione S-transferase, but this is conserved in a number of strains that do not encode an adjacent NRPS so is unlikely to feature as part of this pathway.

NRPS 4: One protein containing a single atypical NRPS module: T-A-AT with predicted phenylglycine or threonine specificity.

NRPS 5: One protein containing a single NRPS module: A-T-TE with predicted threonine specificity.

NRPS 6: One protein containing a single NRPS module: A-T-AT with predicted glutamate specificity. Adjacent to additional putative biosynthetic genes (including genes that encode isochorismitases, peptidases, and glycosyltransferases), but these differ between strains.

NRPS 7: One protein containing a putative NRPS module: sulfotransferase-A-T-reductase with predicted phenylglycine specificity. Homologous gene cluster analysis in antiSMASH indicates a wider gene cluster: methyltransferase, dioxygenase, dehydrogenase, epimerase, oxygenase, transcriptional regulator, transporter, NRPS, dehydrogenase, kinase.

NRPS 8: Single-module PKS: KS-T-TE. Encoded alongside a 4'-pantetheinephosphotransferase and an ABC transporter. These three genes are homologous to a contiguous set of genes (PSF113_3664 to PSF113_3666) within a larger lankacidin-like PKS BGC in *P. fluorescens* F113.

NRPS 9: One protein containing a single NRPS module: A-T-reductase, with predicted isoleucine specificity.

NRPS 10: One protein containing a single NRPS module fused to a P450: A-T-TE-P450. The specificity of the A domain could not be predicted.

NRPS-PKS: BGC with an unknown product identified in Ps843. NRPS-PKS organization: A-T / KS-AT-KR-C / TE. This reflects the order of genes but may not reflect functional order of the synthetase. antiSMASH analysis indicates a larger set of related genes associated with this NRPS-PKS, with multiple homologous gene clusters found in *P. aeruginosa* strains.

## PKSs

Aryl polyenes: Widespread type II PKS gene clusters detected by antiSMASH, as defined by MIBiG entry BGC0000837 (APE Vf from *A. fischeri* ES114) (*Cimermancic et al., 2014*). Cluster defined as aryl polyene if set of PKS genes are found alongside homologues of other characteristic aryl polyene biosynthetic genes, including ammonia lyase and acyltransferase.

PKS 1: BGC with an unknown product identified in Ps664. Putative type I PKS organization: A / T / KS-AT-KR-T-KS-AT-KR-T / KS-AfsA domain / TE. The AfsA domain has homology to the AfsA family of proteins required for A-factor biosynthesis (*Kato et al., 2007*).

PKS 2: BGC with an unknown product identified in Ps691. Putative type I PKS organization: KS-AT-DH-ER-KR-T. This single PKS module is associated with polysaccharide biosynthesis proteins, which is conserved across homologous clusters identified from antiSMASH analysis.

PKS 3: BGC with an unknown product identified in multiple strains. Putative type III PKS gene cluster encoding the following conserved proteins: type III PKS (NCBI conserved domain cd00831), oxidoreductase, SnoaL-like protein, methyltransferase, methyltransferase.

## Siderophores

Pyoverdine: Characteristic biosynthetic genes identified by antiSMASH. Analysis complicated as genes are usually distributed between distinct genomic loci (*Gross and Loper, 2009*).

Achromobactin: Siderophore produced by *P. syringae* pv. *syringae* B728a (*Berti and Thomas, 2009*). Gene cluster defined as achromobactin when homologues of all biosynthetic genes (Psyr2582-Psyr2589) are present in a single BGC.

Pseudomonine: Siderophore produced by *P. fluorescens* WCS374 (*Mercado-Blanco et al., 2001*). Gene cluster defined as pseudomonine when homologues of all biosynthetic genes (as defined in MIBiG entry BGC0000410) are present in a single BGC.

Ornicorrugatin-like: Lipopeptide siderophore produced by *P. fluorescens* AF76 (*Matthijs et al., 2008*), whose gene cluster is described in *P. fluorescens* SBW25 (*Cheng et al., 2013*). Gene cluster defined as ornicorrugatin-like when homologues of all SBW25 genes are present in a single BGC.

Pyochelin-like 1: Gene cluster is similar to the gene cluster to pyochelin (*Patel and Walsh, 2001*) but contains an extra NRPS module predicted to incorporate cysteine (*Appendix 1—figure 1*). NRPS domain organization: A / C-A-T-C-A-T / C-A-MT-T-TE; predicted specificity = dihydroxybenzoic acid-Cys-Cys-Cys. This would be the correct organization for an ulbactin F-like molecule (*Igarashi et al., 2016*).

Pyochelin-like 2: Similar NRPS organization to 'pyochelin-like 1' gene cluster, but different set of associated genes (*Appendix 1—figure 1*). NRPS domain organization: A / C-A-T-C-A / T / C-A-MT-T-TE; predicted specificity = dihydroxybenzoic acid-Cys-Cys-Cys.

Pyochelin-like 3: Canonical pyochelin gene cluster with identical NRPS module organization to the characterized PchDEF system (*Patel and Walsh, 2001*; *Appendix 1—figure 1*): A / T-C-A-T / C-A-MT-T-E; predicted specificity = dihydroxybenzoic acid-Cys-Cys.

Quinolobactin: Siderophore produced by *P. fluorescens* ATCC 17400 (*Matthijs et al., 2004*). Gene cluster defined as quinolobactin when homologues of all biosynthetic genes (as defined in MIBiG entry BGC0000925) are present in a single BGC.

Putative siderophore 1 and 2: Gene clusters encoding pathways predicted to biosynthesize siderophores (*Challis, 2005*). Two distinct gene clusters were identified, so are defined as 'Putative siderophore 1' (encoding: diaminopimelate decarboxylase, PLP-dependent enzyme, dehydrogenase, IucA/IucC-like siderophore biosynthesis protein, major facilitator transporter, aminotransferase) and 'Putative siderophore 2' (encoding: argininosuccinate lyase-like protein, IucA/IucC-like siderophore biosynthesis protein, major facilitator transporter, IucA/IucC-like siderophore biosynthesis protein, diaminopimelate decarboxylase). A few strains encoded IucA/iucC-like proteins, but no other siderophore biosynthesis proteins were encoded alongside these proteins, so were not annotated as siderophore BGCs.

## Ribosomally synthesized and post-translationally modified peptides (RiPPs)

Microcin B17-like: BGCs with homology to the microcin B17 gene cluster from *E. coli* (*Ghilarov et al., 2019*; *San Millán et al., 1985*), encoding the following proteins: McbA-like precursor peptide, McbB-like cyclodehydratase component (TIGR04424), McbC-like flavin-dependent dehydrogenase, McbD-like YcaO domain protein, McbE-like immunity protein, McbF-like immunity protein. Unlike the microcin B17-like gene clusters identified in *P. syringae* (*Metelev et al., 2013*), no McbG homologues were encoded in any of the strains in this current study.

Lanthipeptide: Putative precursor peptide encoded alongside a lanthipeptide synthetase (LanM family, TIGR03897).

YcaO cluster: Putative BGC encoding YcaO and TfuA domain proteins, which are characteristic of RiPP biosynthesis (*Santos-Aberturas et al., 2019*). BGC organization: putative precursor peptide, YcaO domain protein, TfuA domain protein, unknown domain protein, ubiquitin-like domain protein, E1/ThiF-like domain protein, major facilitator superfamily transporter, methyltransferase.

'Pep' BGCs: All 'Pep' BGCs encode short peptides alongside DUF692 domain proteins, which have been shown to be essential for the biosynthesis of the RiPP methanobactin (*Kenney et al., 2018*). Below are lists of proteins encoded in each putative BGC (*Appendix 1—figure 2*):

Pep1: Short peptide (DUF2282), DUF692 protein, DUF2063 protein, DoxX domain protein (pfam07681).

Pep2: Short peptide (DUF2282), DUF692 protein, DUF2063 protein, methyltransferase, cardiolipin synthase.

Pep3: Short peptide (COG3767), DUF692 protein, DUF2063 protein, DMT family transporter, LysR family transcriptional regulator.

Pep4: Short peptide (DUF2282), DoxX domain protein, hydrolase, short peptide (no conserved domain), DUF692 protein, DUF2063 protein, hydrolase.

Pep5: short peptide (no conserved domain), DUF692 protein, DUF2063 protein, DoxX domain protein.

## Terpenes

Carotenoid: BGC containing homologues of all carotenoid genes defined in MIBiG entry BGC0000642 from *Enterobacteriaceae* bacterium DC413 (*Sedkova et al., 2005*).

Unknown terpene 1: Putative BGC identified in Ps655. Terpene synthase/cyclase (NCBI conserved domain cd00687) encoded alongside a polyprenyl synthetase (pfam00348).

Unknown terpene 2: Putative BGC identified in Ps706. Fused terpene synthase/P450, methyltransferase, isopentenyl diphosphate isomerase.

## Others

HCN: BGC containing homologues of *hcnABC*, which together encode the HCN synthase complex. The *hcn* operon from *P. aeruginosa* PAO1 (PA2193–PA2195, *Appendix 1—figure 2*; *Pessi and Haas, 2000*) was used with MultiGeneBlast.

Dimethyl sulfide (DMS): MegL (methionine gamma-lyase) and MddA (methyltransferase) convert methionine to DMS in *Pseudomonas deceptionensis* (*Carrión et al., 2015*). A DMS BGC is defined

when a strain encodes homologues of both MddA and MegL with over 60% identity. A score of 1 was defined when it only encoded a homologue of MddA with over 60% identity but not MegL.

Tabtoxin-like: *Pseudomonas* beta-lactam whose BGC is defined by MIBiG entry BGC0000846 (*Kinscherf and Willis, 2005*). Identified by antiSMASH and confirmed by MultiGeneBlast.

Indole-3-acetic acid 1 (IAA 1): The IAA BGC encodes homologues of IaaM (tryptophan 2-monooxygenase) and IaaH (indoleacetamide hydrolase) from *Pseudomonas savastanoi* (for the indole-3-acetamide pathway to IAA) (*Palm et al., 1989*). Homologues of these were identified using MultiGeneBlast.

Indole-3-acetic acid 2 (IAA 2): Defined as encoding a protein with high homology to IdaA (PSPTO_0092) from *P. syringae* pv. DC3000 (*McClerklin et al., 2018*). All strains encoded proteins with >88% identity to AldA. MultiGeneBlast analysis showed that these genes are all present in exactly the same genetic context as in *P. syringae* pv. DC3000.

Homoserine lactone 1: Defined as encoding a protein homologous to an acyl-homoserine-lactone synthase (pfam00765) from Ps887 (identified by antiSMASH analysis).

Homoserine lactone 2: Defined as encoding a protein homologous to the acyl-homoserine-lactone synthase HdtS from *P. fluorescens* F113 (*Laue et al., 2000*).

A-factor-like: Identified by antiSMASH as an A-factor-like BGC in Ps664. Clustered genes encode an AfsA-like protein (*Kato et al., 2007*), a hydrolase, a P450 and a major facilitator superfamily transporter. This lacks the reductase encoded in a classical A-factor gene cluster (*Kato et al., 2007*).

Aminoglycoside: Identified by antiSMASH as an aminoglycoside-like BGC in Ps639 (*Appendix 1—figure 3*). The BGC encodes the following enzymes that could assemble an aminoglycoside (*Kudo and Eguchi, 2009*) (putative biosynthetic roles are indicated): a 2-deoxy-scyllo-inosose synthase (47% identity to paromomycin homologue, ParC), a phosphoribosyltransferase (33% identity to neomycin homologue, NeoL), a neamine aminotransferase (36% identity to neomycin homologue, NeoN), a 2'-N-acetylparomamine deacetylase (37% identity to neomycin homologue, NeoD), a L-glutamine:scyllo-inosose aminotransferase (56% identity to paromomycin homologue, ParS), a 6'-hydroxyparomomycin dehydrogenase (51% identity to paromomycin homologue, ParQ), a 2-deoxystreptamine N-acetyl-glucosaminyltransferase (44% identity to neomycin homologue, NeoM), and a 2OG-Fe(II) oxygenase that is not homologous to known aminoglycoside biosynthetic enzymes.

Coronamic acid: This unusual amino acid (1-amino-2-ethylcyclopropane carboxylic acid) forms part of the *P. savastanoi* phytotoxin coronatine. BGC identified by antiSMASH in Ps834 contains homologues of all coronamic acid biosynthetic genes, as defined by *Couch et al., 2004* (MIBiG entry BGC0000328). The Ps834 BGC is not associated with polyketide or ligase genes required for full coronatine biosynthesis (*Bown et al., 2017*).

Fosfomycin-like: The fosfomycin BGC from *P. syringae* PB-5123 (*Kim et al., 2012*) was used with MultiGeneBlast to search for similar BGCs. A significant number of homologous proteins are clustered in two strains, including multiple proposed biosynthetic proteins: trans-homoaconitate synthase (Psf2 homologue), epoxidase (Psf4 homologue), 6-phosphogluconate dehydrogenase (Psf3 homologue), fumarylacetoacetate hydrolase, a hypothetical protein, phosphoenolpyruvate phosphomutase (Psf1 homologue) (*Appendix 1—figure 3*).

N-acetylglutaminylglutamine amide (NAGGN): Dipeptide BGC identified by antiSMASH. Representative BGC present in *P. aeruginosa* PAO1 (*Sagot et al., 2010*) (genes PA3459 and PA3460).

Beta-lactones: Identified by antiSMASH and defined by the presence of an AMP-dependent synthetase/ligase and a 2-isopropylmalate synthase (*Robinson et al., 2020*). Three distinct BGC types were identified, where characteristic examples are present in Ps619 (beta-lactone 1), Ps639 (beta-lactone 2), and Ps659 (beta-lactone 3).

Ectoine-like 1: BGC identified by antiSMASH in Ps663, although this is different to the well-characterized *ectABC* cluster identified in *Pseudomonas stutzeri* (*Seip et al., 2011*). The putative Ps663 cluster encodes an AMP-dependent ligase, a hypothetical protein, a dehydrogenase, an ectoine synthase, and a major facilitator superfamily transporter.

Ectoine-like 2: BGC identified by antiSMASH in Ps664 and differs from 'ectoine-like 1' BGC. Features similarities with characterized ectoine gene clusters (e.g., MIBiG entry BGC0000859),

including homologues of the ectoine synthase and transaminase genes. The Ps664 BGC encodes ectoine synthase, transaminase, a dioxygenase, and a putative transporter.

### Pyrroloquinoline quinone (PQQ)

To search for PQQ BGCs, the *P. fluorescens* Pf0-1 PQQ BGC (*Choi et al., 2008*) was used with MultiGeneBlast. Specifically, proteins PqqF, PqqA, PqqB, PqqC, PqqD, PqqE, and PqqM were used, which are encoded by contiguous genes in *P. fluorescens* Pf0-1. This analysis identified a series of distinct PQQ BGC variants (*Appendix 1—figure 4*):

PQQ1: BGC encoding all PQQ proteins as defined above.
PQQ2: BGC encoding PqqF, PqqA, PqqB, PqqC, PqqD, PqqE, but not PqqM. A putative amidase is encoded alongside PqqF, which could potentially functionally replace PqqM.
PQQ3: BGC encoding PqqA, PqqB, PqqC, PqqD, PqqE, but not PqqF or PqqM. BGC is only found in strains that also encode a PQQ1-type BGC.
PQQ4: BGC encoding PqqA, PqqC, PqqD, PqqE, but not PqqB, PqqF or PqqM. BGC is only found in strains that also encode a PQQ1-type BGC.

### Gene clusters not found in this strain collection

In addition to the specific BGCs described above that are found in one or more strains in this study, a number of *Pseudomonas* BGCs were not present in any strains, as described below:

Bicyclomycin: *P. aeruginosa* SCV20265 BGC (*Vior et al., 2018*) was used in MultiGeneBlast.
Cyclodipeptides: tRNA-dependent cyclodipeptide synthases from *Pseudomonas protegens* (NCBI accession OKK65715.1) and *P. aeruginosa* (WP_003158562.1) were used in BLAST analyses.
Pyreudiones: No strains encode the standalone C-A-T-TE NRPS that defines the *P. fluorescens* HKI0770 BGC (*Klapper et al., 2016*).
Kalimantacin: No strains encode the large hybrid PKS-NRPS that makes this molecule (*Uytterhoeven et al., 2016*).
Brabantamide: BraA to BraE proteins from *Pseudomonas* sp. SH-C52 (*Schmidt et al., 2014*) was used with MultiGeneBlast.
Pseudopyronines: The characterized PpyS protein from *P. putida* BW11M1 (*Bauer et al., 2015*) was used in BLAST analyses.
2,4-diacylphloroglucinol (DAPG): The biosynthetic genes (*phlABCD*) from *P. fluorescens* F113 (*Delany et al., 2000*) were used with MultiGeneBlast.
Phenazines: The BGC from *P. fluorescens* 2-79 (*Mavrodi et al., 1998*) (NCBI accession L48616.1) was used with MultiGeneBlast.
Pyrrolnitrin: The BGC (*prnABCD*) from *Pseudomonas aurantiaca* (previously named *P. fluorescens*) BL915 (*Hammer et al., 1997*) was used with MultiGeneBlast.
2,5-Dialkylresorcinols: The *darABC* genes from *Pseudomonas chlororaphis* PCL1606 (*Calderón et al., 2013*) (NCBI accession JQ663992.1) were used with MultiGeneBlast.
Toxoflavin: The toxoflavin BGC from *P. protegens* Pf-5 (PFL_1028 to PFL_1037) (*Philmus et al., 2015*) was used with MultiGeneBlast.
L-2-Amino-4-methoxy-trans-3-butenoic acid (AMB): The *ambABCDE* gene cluster from *P. aeruginosa* PAO1 (PA2302-PA2306) (*Lee et al., 2010*) was used with MultiGeneBlast.

### Accessory genome loci

#### Exopolysaccharides

*psl* operon: The *psl* gene cluster (PA2231 to PA2245) from *P. aeruginosa* PAO1 (*Jackson et al., 2004*) was used with MultiGeneBlast. Score = 2 for all genes present; score = 1 for missing 1–3 genes in the cluster.
*wss* operon: The *wss* gene cluster (*wssA-J*) from *P. fluorescens* SBW25 (*Spiers et al., 2003*) was used with MultiGeneBlast. Score = 2 for all genes present; score = 1 for only homologues of *wssA-E* clustered.
*pel* operon: The *pel* gene cluster (*pelA-G*) from *P. aeruginosa* PA14 (*Friedman and Kolter, 2004*) was used with MultiGeneBlast. Score = 2 for all genes present.
*pga* operon: The *pga* gene cluster (*pgaA-D*) from *P. fluorescens* SBW25 (PFLU0143-PFLU0146) (*Lind et al., 2017*) was used with MultiGeneBlast. Score = 2 for all genes present.

Alginate: The alginate gene cluster from *P. fluorescens* SBW25 (PFLU0979-PFLU0990) (*Maleki et al., 2016*) was used with MultiGeneBlast. Score = 2 for all genes present.

## Stress response polysaccharides

Alpha glucan biosynthesis: Biosynthetic proteins from *P. syringae* pv. DC3000 (PSPTO_2760-62, PSPTO_3125-30, PSPTO_5165) (*Freeman et al., 2010*) were used with BLAST and MultiGeneBlast. Score = 2 for homologues of all genes present; score = 1 when 1–2 genes were absent from the PSPTO_3125-30 cluster.

Trehalose degradation: PSPTO_2952 protein from *P. syringae* pv. DC3000 (NCBI accession NP_792749.1) was used with BLAST. Score = 2 for homologues > 70% identity.

## Lipopolysaccharides

Fuzzy spreader: The *fuzVWXYZ* operon from *P. fluorescens* SBW25 (PFLU0475-PFLU0479) (*Ferguson et al., 2013*) was used with MultiGeneBlast. Score = 2 for all genes present in contiguous cluster. Score = 1 for all genes in a cluster but intercalated with additional genes.

## Proteinaceous adhesins

LapA adhesin: PFL_0133 protein from *P. protegens* Pf-5 (NCBI accession AAY95545.1) (*Boyd et al., 2014*) was used with BLAST. Score = 2 for homologues > 60% identity.

*bapABCD* adhesin: The *bapABCD* gene cluster from *P. aeruginosa* PAO1 (PA1874-PA1877) (*de Bentzmann et al., 2012*) was used with MultiGeneBlast. Score = 2 for all genes present.

Curli fimbriae: The curli fimbriae gene cluster of *P. fluorescens* Pf0-1 (Pfl01_1982-Pfl01_1993) (*Dueholm et al., 2012*) was used with MultiGeneBlast. Score = 2 for all genes present.

## Plant-bacterial communication

Auxin (IAA) catabolism: The IAA catabolism gene cluster (*iacA-I*, NCBI accession EU360594.1) from *P. putida* strain 1290 (*Leveau and Gerards, 2008*) was used with MultiGeneBlast. Clusters containing all catabolic genes but lacking a homologue of the regulatory gene *iacR* were scored as 2; one cluster lacked *iacF* so was scored as 1.

Phenyl acetic acid (PAA) catabolism: The PAA catabolism gene cluster from *P. protegens* Pf-5 (PFL_3128-PFL_3140) (*Teufel et al., 2010*) was used with MultiGeneBlast. Strains containing homologues of all genes were scored as 2. Some strains encoded the proteins across two distinct genomic loci (e.g., Ps673) but all proteins had high homology to *P. protegens* Pf-5 proteins (>70% identity) so were still scored as 2.

1-Aminocyclopropane-1-carboxylate (ACC) deaminase: AcdS from *P. fluorescens* F113 (NCBI accession AEV63500.1) (*Saravanakumar and Samiyappan, 2007*) was used with BLAST. Score = 2 for homologues > 70% identity.

3-Hydroxybutanone (acetoin) catabolism 1: The acetoin catabolism gene cluster (*acoA-C*, *acoX*, *adh*) from *P. protegens* Pf-5 (PFL_2168-PFL_2172) (*Huang et al., 1994*) was used with MultiGeneBlast. Score = 2 for all genes present in cluster. Score = 1 for clusters that lacked a homologue of AcoX.

Acetoin catabolism 2: Acetoin reductase from *P. fluorescens* A506 (NCBI accession AFJ57022.1) was used with BLAST. Score = 2 for homologues > 70% identity.

## Secretion systems

Type II secretion system (T2SS): The T2SS gene cluster (*gspCDEFGHIJKLM*) of *P. fluorescens* SBW25 (PFLU2415-PFLU2425) (*Scales et al., 2015*) was used with MultiGeneBlast. Some genomes contain all genes within the same locus (score = 2), whereas some seem to only contain DEFGH and sometimes one pseudolipin, so are scored 1. Some (e.g., Ps664 and Ps720) have two authentic systems and others have a complete SBW25-like system as well as a smaller DEFGH-like clusters.

Type III secretion system (T3SS): The T3SS gene cluster of *P. fluorescens* SBW25 (PFLU0708, PFLU0710-PFLU0727) was used with MultiGeneBlast. Score = 2 when homologues of all genes were found in a gene cluster. A score of 2 was also provided if homologues of the effector (PFLU0708) were not encoded in the gene cluster. Some gene clusters feature additional genes that could reflect additional components of the secretion system absent from SBW25. For example, the gene cluster in Ps664 included a type III secretion ATP synthase (HrcN), a hypothetical protein, and type III secretion protein (HrcV).

Type VI secretion system (T6SS): The T6SS gene cluster of *P. fluorescens* A506 (PflA506_2406-PflA506_2421) was used with MultiGeneBlast. Score = 2 when homologues of all genes were found in a gene cluster. Some gene clusters feature additional genes that could reflect additional components of the secretion system absent from A506.

## Exoenzymes

Pectin lyase 1: Protein PFLU2293 from *P. fluorescens* SBW25 was used in BLAST analysis. Clear distinction between high identity (>90%, score = 2) and low identity (<30%, score = 0) proteins.

Pectin lyase 2: Protein PFLU2269 from *P. fluorescens* SBW25 was used in BLAST analysis. Score = 2 with >80% identity.

Pectate lyase: Protein PFLU3229 from *P. fluorescens* SBW25 was used in BLAST analysis. Score = 2 with >80% identity. Strains such as Ps706 have high homology across to have 'split' proteins and were scored as 1.

Chitinase ChiC: Protein PFL_2091 from *P. protegens* Pf-5 was used in BLAST analysis. Score = 2 with >50% identity. This cutoff retained proteins annotated as 'Chitinase D.'

Chitinase class 1: Protein PSF113_1189 from *P. fluorescens* F113 was used in BLAST analysis. Score = 2 with >60% identity. This cutoff retained proteins annotated as 'Chitinase class I.'

Extracellular alkaline metalloprotease AprA: Protein PFL_3210 from *P. protegens* Pf-5 was used in BLAST analysis. Score = 2 with >80% identity; score = 1 with >65% identity.

LipA lipase: Protein PFLU0569 from *P. fluorescens* SBW25 was used in BLAST analysis. Score = 2 with >75% identity.

LipB lipase: Protein PFLU3141 from *P. fluorescens* SBW25 was used in BLAST analysis. Score = 2 with >80% identity; score = 1 with >65% identity.

## Toxins

Insecticidal toxin complex (Tc) gene clusters: To search for putative insecticidal toxin complex (Tc) gene clusters, the examples reported by *Rangel et al., 2016* were used with MultiGeneBlast and/or BLAST analysis. Specifically, the following proteins were used:

Type I = *P. chlororaphis* 30-84 (Pchl3084_2947 and Pchl3084_2950)
Type II = *P. fluorescens* Q2-87 (PflQ2_0667-0670)
Type III = *P. fluorescens* Q8r1-96 (PflQ8_4696, PflQ8_4570-4571 and PflQ8_4580-4581)
Type IV = *P. fluorescens* Pf0-1 (Pfl01_0947-0948 and Pfl01_4453-4456)
Type V = *P. fluorescens* A506 (PflA506_3065-3068)
Type VI = *P. synxantha* BG33R (PseBG33_3799-3804)
Type II/V/VI Tc proteins provided comparable hits (due to sequence homology between components of these toxin systems) whose genes were arranged differently to the characterized examples. These were therefore grouped as a single genotype and scored as 1.

HicAB toxin-antitoxin: The HicAB proteins from *P. aeruginosa* PA1 (*PA1S_06925* and *PA1S_06920*) (*Li et al., 2016*) was used with MultiGeneBlast.

## Accessory genome loci not found in this strain collection

Cytokinin: The cytokinin isopentenyl transferase Ptz from *P. savastanoi* (NCBI accession P06619.1) (*Powell and Morris, 1986*) was used in a BLAST analysis.

CdrA adhesin: Protein PA4625 from *P. aeruginosa* PAO1 (*Borlee et al., 2010*) was used in a BLAST analysis.

N-acyl homoserine lactonase: Protein BW979_RS17690 from *Pseudomonas* sp. A214 used in a BLAST analysis. This provided some hits, but all were <25% identity so were scored as 0.

Fit toxin: The Fit insect toxin cluster of *P. protegens* Pf-5 (PFL_2980 to PFL_2987) (*Péchy-Tarr et al., 2008*) was used with MultiGeneBlast.

Insecticidal protein IPD072Aa: Protein IPD072Aa from *P. chlororaphis* (NCBI accession KT795291.1) (*Schellenberger et al., 2016*) was used in a BLAST analysis.

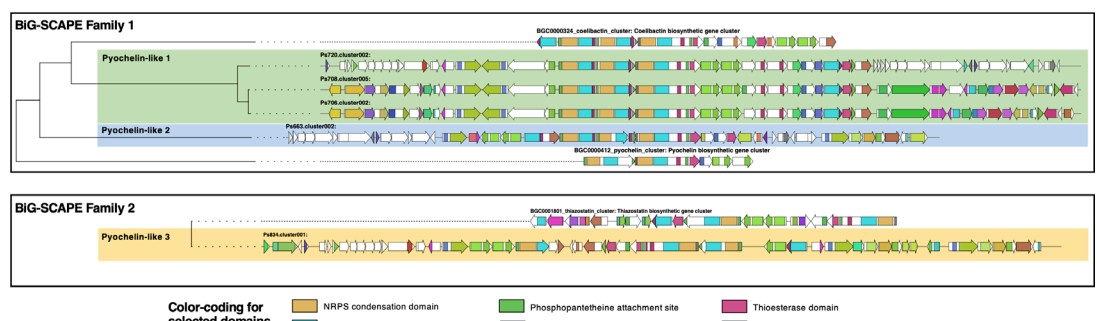

**Appendix 1—figure 1.** Comparison of characterized pyochelin-like biosynthetic gene cluster (BGCs) versus pyochelin-like BGCs identified in this work. BGCs were retrieved from MIBiG (https://mibig.secondarymetabolites.org) (*Kautsar et al., 2020*) for pyochelin (BGC0000412), coelibactin (BGC0000324), and thiazostatin (BGC0001801). BGC homology was assessed using BiG-SCAPE (https://git.wageningenur.nl/medema-group/BiG-SCAPE/) (*Navarro-Muñoz et al., 2020*) in glocal mode with a cutoff of 0.75. Two figures show two distinct families.

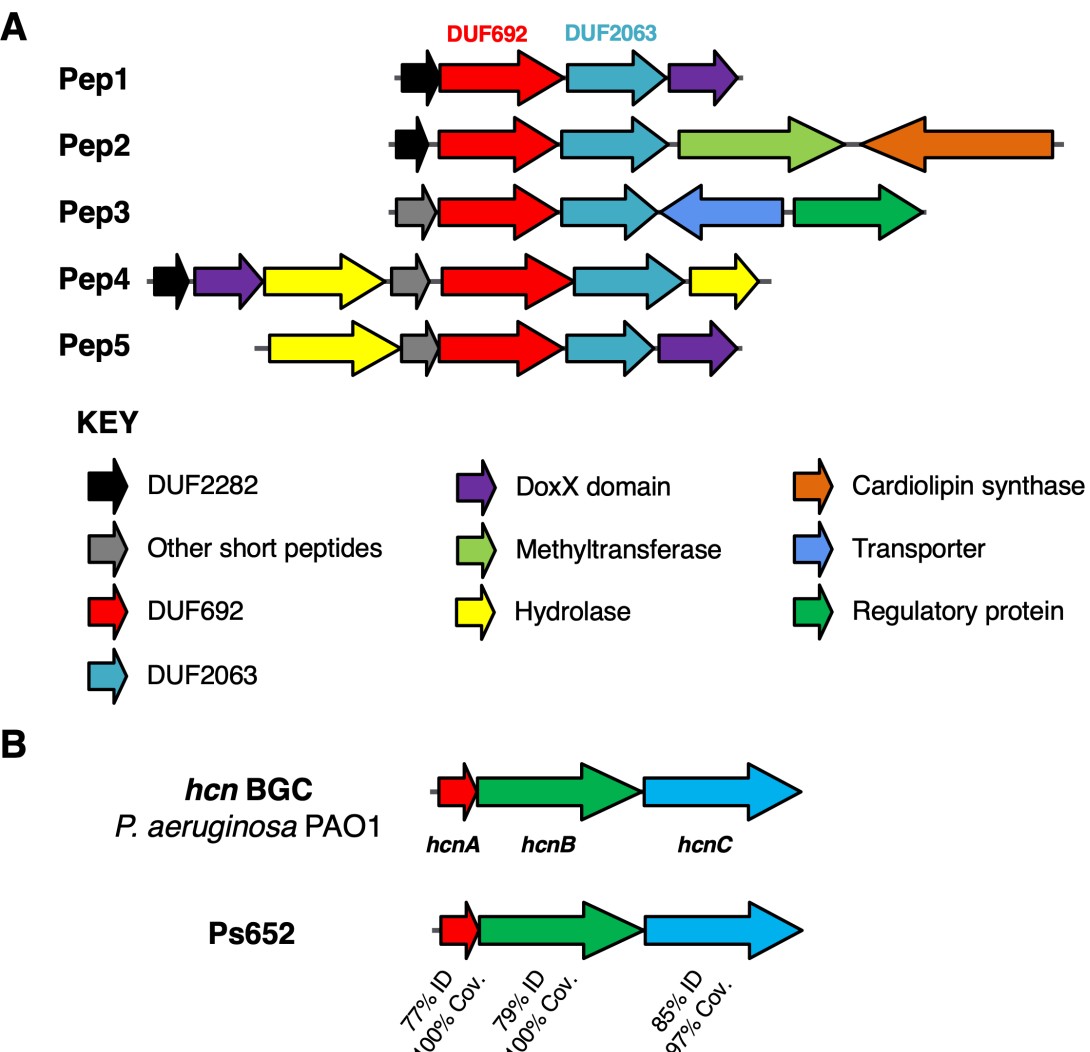

**Appendix 1—figure 2.** Biosynthetic gene clusters (BGCs) that correlate with *S. scabies* inhibition. (**A**) Putative Pep BGCs. (**B**) Hydrogen cyanide (HCN) BGC compared to the characterized *P. aeruginosa* PAO1 BGC (*Pessi and Haas, 2000*).

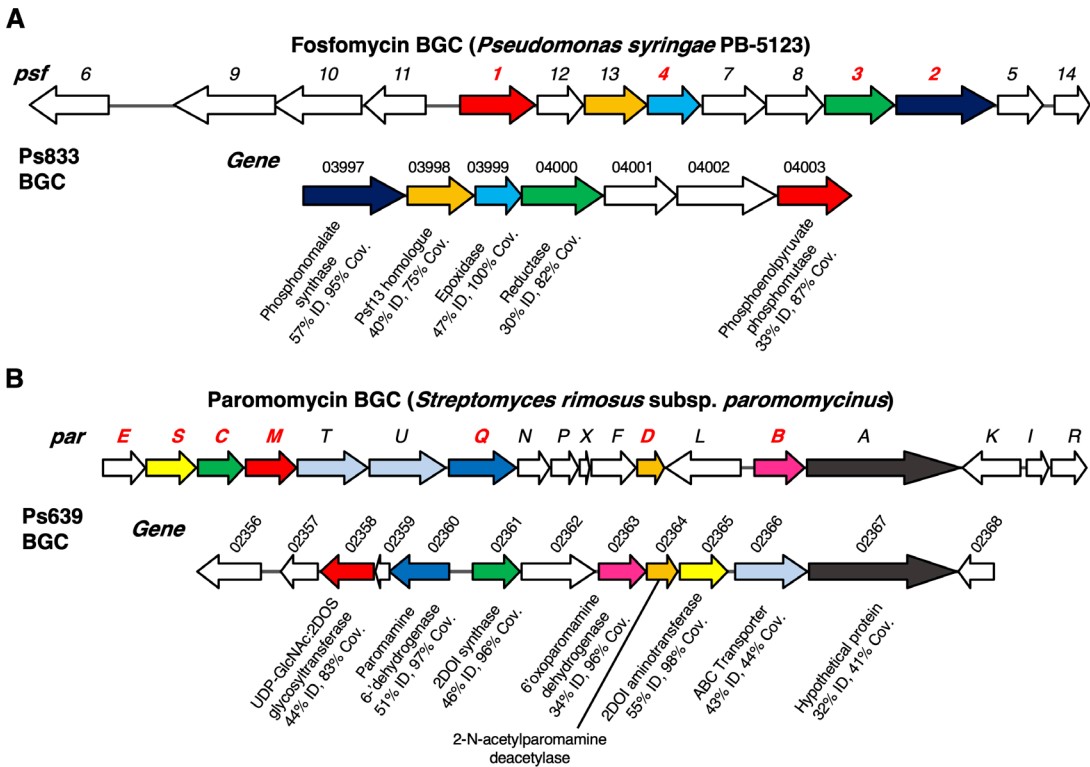

**Appendix 1—figure 3.** Examples of biosynthetic gene clusters (BGCs) not previously characterized in *P. fluorescens*. (**A**) Fosfomycin-like BGC in strain Ps833. Comparison to the *psf* BGC in *P. syringae* PB-5123 is shown, where color coding represents homologous genes (identity and coverage values relate to encoded proteins). Gene numbers colored in red have been experimentally characterized and encode enzymes that catalyze key biosynthetic steps (***Kim et al., 2012***; ***Olivares et al., 2017***). (**B**) Aminoglycoside-like BGC in strain Ps639. Comparison to the *par* BGC in *S. rimosus* is shown, where color coding represents homologous genes (identity and coverage values relate to encoded proteins). Gene numbers colored in red are predicted to be required for the biosynthesis of the minimal aminoglycoside, neamine (***Kudo and Eguchi, 2009***).

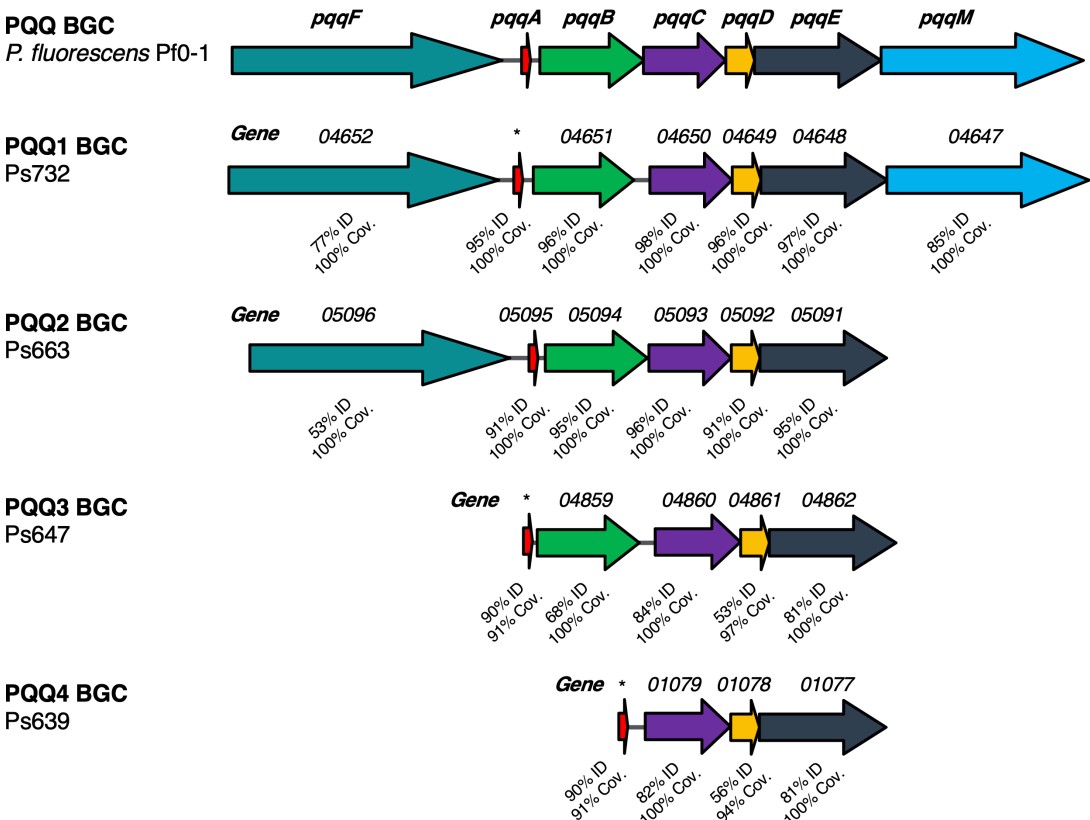

**Appendix 1—figure 4.** Different pyrroloquinoline quinone (PQQ) biosynthetic gene clusters (BGCs) identified in this study. Comparison to the *pqq* BGC in *P. fluorescens* Pf0-1 is shown, where color coding represents homologous genes and % identity/coverage indicate how similar the encoded proteins are to the Pf0-1 PQQ proteins. *Gene not annotated but *pqqA* homologue identified by tblastn analysis.

# Appendix 2

## Genotypes and phenotypes that correlate with the suppression of different plant pathogens

On-plate assays of the *Pseudomonas* field isolates with *P. infestans* (oomycete that causes potato blight) and *G. graminis* var. *tritici* (fungus that causes take-all disease of cereal crops) revealed strong positive correlations between suppressive phenotypes for each of the pathogens tested, such as a correlation coefficient of $\rho$ = 0.55 between *S. scabies* and *P. infestans* inhibition (*Figure 3—figure supplement 1*, *Appendix 2—figure 1A*). More generally, multiple *Pseudomonas* genotypes that correlated with suppression of *S. scabies* also correlated with suppression of the other pathogens (*Appendix 2—figure 1A*). The roles of HCN and CLPs in this inhibitory activity were assessed by testing Ps619 and Ps682 mutants towards *P. infestans* and *G. graminis* (*Appendix 2—figure 1*).

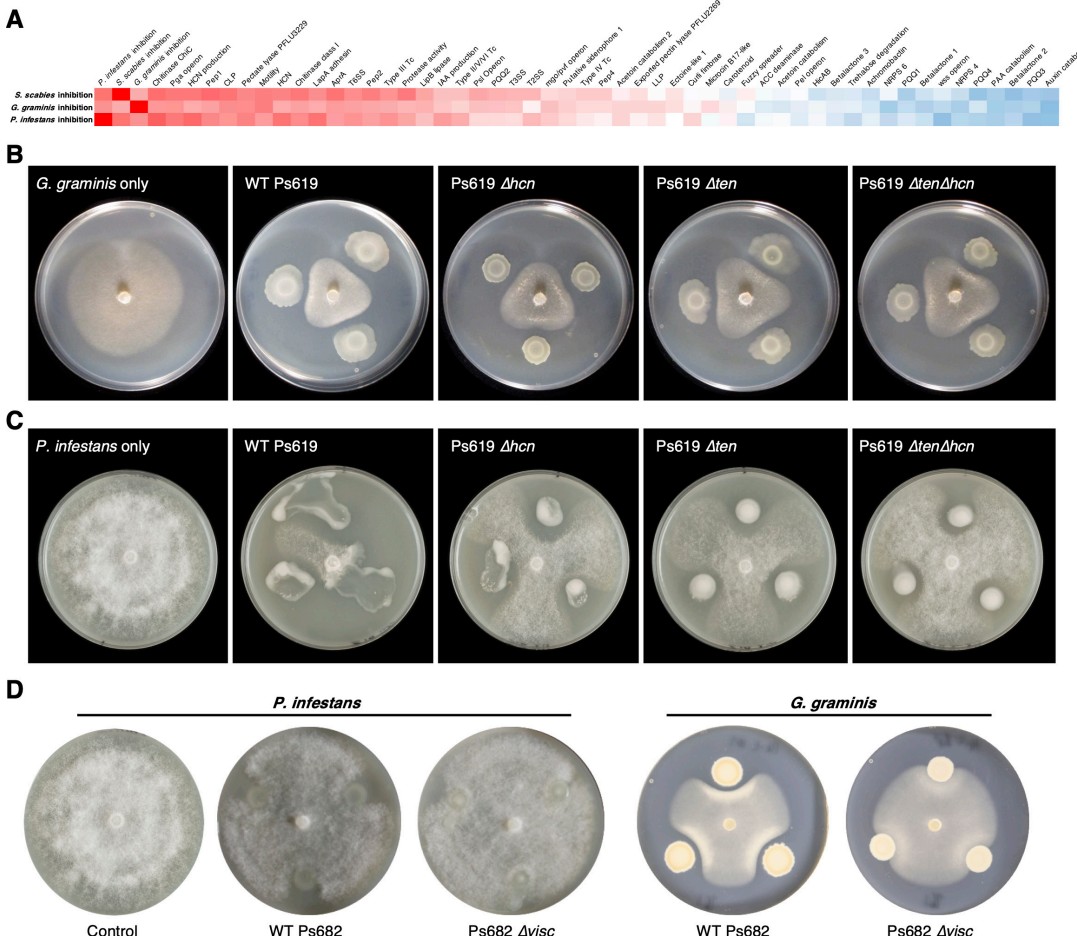

**Appendix 2—figure 1.** Genotypes and phenotypes that correlate with the suppression of different plant pathogens. (**A**) Heatmap of Pearson correlation coefficients of pathogen inhibition versus genotypes and phenotypes (see *Figure 3—figure supplement 1* for full correlations; the same color scale is used). (**B**) On-plate inhibition of *G. graminis* growth by wild-type (WT) and mutant Ps619 strains. (**C**) On-plate inhibition of *P. infestans* growth by WT and mutant Ps619 strains. (**D**) WT and mutant Ps682 activity towards *P. infestans* and *G. graminis*.

