## [Editor Report]

This work uses large-scale genome sequencing and analysis, mass spectrometry, and bioassays to investigate the genomic diversity of *Pseudomonas* strains and their role in plant protection. The authors identified key metabolites that inhibit *Streptomyces scabies*, the causal agent of potato scab, and showed how genomic diversity in closely related bacterial strains can contribute to plant pathogen suppression in the field.

---

## [Decision Letter]

**Decision letter after peer review:**

[Editors’ note: the authors submitted for reconsideration following the decision after peer review. What follows is the decision letter after the first round of review.]

Thank you for submitting your work entitled "Pan-genome analysis identifies intersecting roles for *Pseudomonas* specialized metabolites in potato pathogen inhibition" for consideration by *eLife*. Your article has been reviewed by 3 peer reviewers, and the evaluation has been overseen by a Reviewing Editor and a Senior Editor. The reviewers have opted to remain anonymous.

Our decision has been reached after consultation between the reviewers. Based on these discussions and the individual reviews below, we regret to inform you that your work will not be considered further for publication in *eLife*.

While there was enthusiasm for several aspects of the study, this enthusiasm was tempered by the opinion that the take-home message of the paper was unsatisfying. The consensus opinion is that helpful insights could likely be gained from in planta work to bridge the gap between in vitro and field data, allowing for a clearer message. We are therefore returning this study to you with encouragement to take the time to attempt these type of clarifying in vivo experiments, and to resubmit to *eLife* once they are complete. To give you ample time to take this course, should you choose to do so, we are formally rejecting the paper to remove any time pressure. We hope you will find the following reviews constructive as you reassess the study, and we hope to see a significantly revised manuscript (complete with new experiments) returned to *eLife* as a new submission in the future.

*Reviewer #1:*

This paper encompasses a substantial amount of work on an important and exceedingly relevant topic of crop biocontrol under various aridity conditions. The authors address the potato scab disease caused by *Streptomyces* scabie, a disease currently best managed by irrigation. The authors suspect that the protection under irrigation is mediated by the rise of specific pathogen suppressive bacteria in the soil/rhizosphere. The authors chose to focus on Pseudomonads which they cultivate from the field and assay them phenotypically as well as sequence and compare their genomes, identifying differentially distributed BGCs that correlate with pathogen suppression, in vitro. They explore the roles two classes of metabolites (HCN and CLPs) in inhibiting Ss. They ultimately find that Ss suppressive strains (in vitro) are not enriched in the protected irrigated fields.

Overall, the manuscript is well-written, the figures are clear and beautiful, and the quality of work is noteworthy. However, the story presented is cumbersome, with 8 main figures (and 19S figures!) the paper is not accessible as it should be. More problematic is the lack of in-vivo work (plant or field) and ultimately the discrepancy between the hypothesis and results (suppressive strains are not enriched in irrigated fields).

Unless I missed the point (and I would gladly read an appeal by the authors or other reviewers if I did), the storyline doesn't lead to a logical conclusion. The authors show multiple experiments of notable quality, but I find it hard to describe exactly what the authors found in relation to their stated problem. Were they able to provide insight into the irrigated vs. unirrigated phenomena? Was the in-vitro phenotyping relevant to the field?

Unfortunately, while I find the paper interesting, it isn't currently clear to me that the authors made a discovery that would merit publication in *eLife*. I urge the authors to try and distill the discoveries made in this study into a more concise and logically appealing report.

1. The authors took a genomic approach alongside an array of classical microbiological techniques to study the potential interactions between Pseudomonads and *Streptomyces* scabie under defined laboratory conditions. They reached the conclusion that CLPs and HCN can help Pseudomonads to inhibit and outcompete S. scabie. However, while these results provide well-demonstrated phenotypes, considering the large body of work on Pseudomonad-dependent biocontrol it is not surprising that toxic metabolites like HCN can inhibit competitor growth. While CLPs provide a novel result, the authors do not isolate the effect of CLPs as compared with motility effects – for example, they do not show that the purified CLP can cause these effects.

2. The authors choose to focus on Pseudomonads, which have been already extensively implicated in biocontrol in the last 3 decades, with many studies linking various secondary metabolites to disease control in various crops, mainly in the context of fungal infections, as well as in the context of aridity and irrigation: the authors are should be aware of the multiple papers from the Thomashow's lab on irrigations and biocontrol by Pseudomonads, which are not citated here… (Mavrodi 2012 and 2018 for example).

3. Since all of the phenotypic work in the study is done on agar plates and not on lab grown plants or in the field its completely feasible that the inferred interactions do not take place outside of the lab. This is a major limitation of the current study and a reasonable explanation to their later contradictory findings.

4. Ultimately, using both metagenomics and culturing the authors find no substantial difference in the microbial communities of irrigated vs. unirrigated fields. Thus, the protective effects of irrigation remain a mystery and Pseudomonads have again been shown to inhibit pathogens in-vitro. There could be many possible reasons for this discrepancy. Possibly another suppressive type of bacteria or fungi or maybe just the abiotic conditions imposed under irrigation (lower oxygen levels for example) shape the behavior of the pathogen itself.

*Reviewer #2:*

The manuscript by Stefanato et al. constitutes a brilliant multidisciplinary study of the pan-genomic variation in molecular interactions with plant pathogens, which sheds light on their potential plant-protective role within plant microbiomes. The research includes large-scale genome sequencing and analysis, mass spectrometry, in vitro bioassays and ecological analysis. The discovery that cyclic lipopeptides inhibit bacterial plant pathogens is very novel and exciting, and the general approach provides a blueprint for future pan-genomic analyses of host-microbe interactions, also outside the plant microbiome field. In general, I am of the opinion that this work is very suitable for publication in *eLife*.

However, I do have a number of concerns that I would like to see addressed:

– The quality of the phylogenetic tree used as reference for the analysis is quite important for the rest of the analysis. Given that single marker genes were used, is there a specific reason why the authors chose to use FastTree instead of a maximum likelihood algorithm like IQTree or RAxML(-NG)? Unlike the authors mention in the methods, FastTree itself is not a maximum likelihood algorithm but an 'approximate maximum likelihood' algorithm. Also, it would be very useful if bootstrap information would be provided, at least somewhere in the supplement or in an online data file.

– The authors note that they find some BGCs that are 'rarely found in pseudomonads'. Given that draft genomes were used, did the authors make sure that there was no contamination of the genome assemblies by co-isolated strains? A scary amount of draft genomes in the public databases contain contigs from co-isolates or contaminant strains. A simple CheckM analysis (https://genome.cshlp.org/content/25/7/1043.full) would suffice to rule this out.

– With regard to the correlation analysis, it should be noted that this can be highly biased by the phylogenetic structure of the data, as well as isolation bias (having isolated more closely related strains from a taxon leads to higher correlations to elements specific to that taxon). While this is an inherent limitation, it should be kept in mind. Specifically, I believe that the Chi-squared test (mentioned in Figure 3) is, strictly speaking, not appropriate, as it assumes independence of the observations, which is not the case for gene clusters found in strains that are related to each other through recent common ancestry. Regardless, the calculations are, in practice, still able to identify the patterns that are also visible simply by eye in Figure 2, but the statistics may not be so relevant.

– For the new viscosin and tensin analogues, it is not clearly indicated in the text/figures how they are related to known viscosins/tensins. 'Viscosin-like' seems to imply that it concerns a new variant, but the supplementary figure suggests otherwise.

– The authors conclude that a subset of the pseudomonads function as generalist pathogen suppressors, based on in vitro experiments, and also imply that pseudomonads use CLPs and HCN to inhibit pathogenic streptomycetes. However, it should be noted that in vitro activities may not be representative of in vivo ecological functions, as, in planta, the required molecular triggers to induce the expression of the relevant gene clusters may not be present (in sufficient amounts). I.e., it might well be that in planta, the BGCs are only actively inhibiting a smaller subset of the pathogens than the ones against which their products are active in vitro. I believe this caveat should be clearly indicated in the Discussion section.

*Reviewer #3:*

Overall, this is a very interesting study where integrated genomics, metabolomics, phenotypic analysis and molecular biology approaches were used to identify the genetic determinants of *Pseudomonas* antagonism towards Stretomyces scabies causing the common scab disease of potato. A comparative approach was used where *Pseudomonas* associated with an irrigated field (where less symptoms of common scab are generally observed) and a non-irrigated field (where more disease pressure is observed) were compared. The amount of work invested here is important, the analyses are sound and the paper is overall well written.

Main concerns:

1) The paper (including the title) should more focused on common scab, while the other two pathogens used in parallel should be more accessory (validation tools only).

2) The introduction should be significantly improved (and more accurate) to better reflect the recent literature on *Pseudomonas* against S. scabies and also on the genomics of phytobeneficial *Pseudomonas*. For example, *Pseudomonas* have been successfully used under soil (Arseneault A, C Goyer, M Filion. 2016. Biocontrol of potato common scab is associated with high *Pseudomonas* fluorescens LBUM223 populations and phenazine-1-carboxylic acid biosynthetic transcripts accumulation in the potato geocaulosphere. Phytopathology 106: 963-970.) and field conditions to control common scab (Arseneault A, C Goyer, M Filion. 2015. *Pseudomonas* fluorescens LBUM223 increases potato yield and reduces common scab symptoms in the field. Phytopathology 105: 1311-1317.). Also, I invite the authors to consult a recent paper on the genomics of phytobeneficial Pseudomomas (Biessy A, A Novinscak, J Blom, G Léger, LS Thomashow, FM Cazorla, D Josic, M Filion. 2019. Diversity of phytobeneficial traits revealed by whole-genome analysis of worldwide-isolated phenazine-producing *Pseudomonas* spp. Environmental Microbiology. 21: 437-455.), which provides valuable information that will guide the authors to modify their introduction section. The authors seem to be unaware of a significant portion of the literature important for their study.

3) It is not clear to me as to how the screening of all the isolated *Pseudomonas* was performed to avoid redundancy. The authors claim that 240 strains of *Pseudomonas* were isolated. Does this mean that these 240 strains are different or that some of these strains are in fact identical. Less than a third of these strains were sequenced, what about the others?

4) As the main conclusion seems to be that the *Pseudomonas* capable of producing CLP and hydrogene cyanide are the most antagnostic against S. scabies, why not perform an in planta experiment showing that at least one wildtype hydrogence cyanide and CLP producer can reduce common scab symptoms (while isogenic mutants for these genes are less efficient). It is surprising to see that these organisms are less present in the irrigated field where less disease is observed. An in planta confirmation would significantly strengthen the conclusion of this study.

[Editors’ note: further revisions were suggested prior to acceptance, as described below.]

Thank you for submitting your article "Pan-genome analysis identifies intersecting roles for *Pseudomonas* specialized metabolites in potato pathogen inhibition" for consideration by *eLife*. Your article has been reviewed by 3 peer reviewers, and the evaluation has been overseen by a Reviewing Editor and Gisela Storz as the Senior Editor. The reviewers have opted to remain anonymous.

The extensive additional work and in planta experiments have addressed previous concerns and greatly improved the work. While the reviewers are enthusiastic about the current version, the authors should still address the following suggestions, raised by the third reviewer, in order to improve the clarity of the manuscript prior to publication.

Essential revisions:

Suggestions to improve clarity and readability.

1. There is growing body of literature showing that drought enriches for commensal *Streptomyces* (and dramatically depletes Proteobacteria) in the rhizosphere, which is reversible after irrigation (i.e. https://doi.org/10.1073/pnas.1717308115; https://doi.org/10.1038/s41477-021-00967-1) (2021). Although these studies are from the perspective of changes that are protective against drought, it seems the same mechanism that results in enrichment of *Streptomyces* (and depletion of Proteobacteria) during drought may deplete *Streptomyces* (and enrich *Pseudomonas*) during irrigation. It would be useful for the authors to compare the changes they see after irrigation to those induced by drought in the literature.

2. I am not sure that the data that led to the conclusions that "irrigation led to a decrease in the proportion of suppressive pseudomonads on potato roots" (568-9) is really rigorous enough to support this counter intuitive conclusion. The sample sizes of suppressive isolates is quite small, and it is hard to be certain that culturing really represents a random and representative sample of what is in the soil. In retrospect, it seems like it would have been more straightforward to perform qPCR to look at abundance of genes or transcripts related to HCN or CLPs in the two soil types. However, as the authors found an increase in the total abundance of *Pseudomonas* after irrigation (Figure 1C), I think they could make a stronger case based on my point above and speculate that an increase in total *Pseudomonas* might contribute to suppression, rather than a change in function of *Pseudomonas* (which is not really supported by the data).

3. For Figure 2, it would be helpful if the authors could label each *Pseudomonas* subgroup as in Garrido-Sanz et al. and/or include some widely studied or type strains to help orient the reader and provide context for the strains and tree. This would also help others in the field use this as a resource to identify functional genes that are *Pseudomonas*-clade specific.

4. For the paragraphs starting on line 192 (mostly related to Figure 2) please ensure you reference the figure or data in each instance. For instance, "Multiple BGCs were commonly found across the sequenced strains, including BGCs predicted to make CLPs (42), arylpolyenes (43) and HCN (38)." This is shown in Figure 2, but it is not referenced in the text so it was hard for me find.

5. There are multiple auxin biosynthesis pathways in *Pseudomonas* (IoaX/IAN, IAH, etc.; for instance, see https://journals.plos.org/plospathogens/article?id=10.1371/journal.ppat.1006811) so it would be helpful to know if the genes identified are all the same pathway. I'm asking in part because it's a bit surprising that IAA biosynthesis is so uncommon in your analysis while other studies have found it to be quite prevalent (for instance, while this study is limited to P. brassicearum, they identified IAA biosynthesis in the whole clade https://www.microbiologyresearch.org/content/journal/jmm/10.1099/jmm.0.001145)

*Reviewer #1 (Recommendations for the authors):*

The authors have made multiple significant improvements to their original work, including in planta and purified CLP experiments, which fully addressed my previous main concerns. I expect this paper will be well appreciated in the field and strongly support its publication in *eLife*.

*Reviewer #2 (Recommendations for the authors):*

The authors did a lot of additional work to improve the paper, including biocontrol assays, genetic analysis and more detailed chemical analysis of the cyclic lipopeptides. The data look solid, and the paper is much improved. I have no remaining objections and am very happy with this new version.

*Reviewer #3 (Recommendations for the authors):*

I have read through the previous reviews and responses and believe the authors have addressed the majority of comments and I think the manuscript is suitable for publication in its current form. My comments and suggestions are mostly focused on improving readability. While there are always more experiments that can be done, this is already an extremely extensive study.

---

## [Author Response]

While there was enthusiasm for several aspects of the study, this enthusiasm was tempered by the opinion that the take-home message of the paper was unsatisfying. The consensus opinion is that helpful insights could likely be gained from in planta work to bridge the gap between in vitro and field data, allowing for a clearer message. We are therefore returning this study to you with encouragement to take the time to attempt these type of clarifying in vivo experiments, and to resubmit to eLife once they are complete. To give you ample time to take this course, should you choose to do so, we are formally rejecting the paper to remove any time pressure. We hope you will find the following reviews constructive as you reassess the study, and we hope to see a significantly revised manuscript (complete with new experiments) returned to eLife as a new submission in the future.

We agree that the lack of key in planta data was a limitation of the original paper. We therefore conducted a 16-week potato scab infection trial where the biocontrol properties of wild type and mutant *Pseudomonas* strains were assessed. This provided two important results:

1. A suppressive strain studied in detail in the manuscript, Ps619, was able to significantly suppress potato scab in planta, thereby supporting the on-plate data reported in the original manuscript.

2. Biocontrol assays with Ps619 *∆ten* mutants demonstrated that cyclic lipopeptide production is a key determinant of Ps619 biocontrol of potato scab, which provides a clear bridge to the results observed for mutants reported in the original manuscript.

The same results were provided by an independent replica experiment; in each case the biocontrol effect was statistically significant. These results are incorporated as Figure 7 in the revised manuscript.

We have also included some key chemical data that was recommended by Reviewer 1, where we have purified and structurally characterised a cyclic lipopeptide from Ps682. We show that the isolated molecule (“viscosin I”) has inhibitory activity towards *Streptomyces scabies*.

Reviewer #1:This paper encompasses a substantial amount of work on an important and exceedingly relevant topic of crop biocontrol under various aridity conditions. The authors address the potato scab disease caused by Streptomyces scabie, a disease currently best managed by irrigation. The authors suspect that the protection under irrigation is mediated by the rise of specific pathogen suppressive bacteria in the soil/rhizosphere. The authors chose to focus on Pseudomonads which they cultivate from the field and assay them phenotypically as well as sequence and compare their genomes, identifying differentially distributed BGCs that correlate with pathogen suppression, in vitro. They explore the roles two classes of metabolites (HCN and CLPs) in inhibiting Ss. They ultimately find that Ss suppressive strains (in vitro) are not enriched in the protected irrigated fields.Overall, the manuscript is well-written, the figures are clear and beautiful, and the quality of work is noteworthy. However, the story presented is cumbersome, with 8 main figures (and 19S figures!) the paper is not accessible as it should be. More problematic is the lack of in-vivo work (plant or field) and ultimately the discrepancy between the hypothesis and results (suppressive strains are not enriched in irrigated fields).

As described in the response to the editor, we have now obtained significant in planta data that ties together much of the genotypic and phenotypic data reported in the original manuscript. We cannot entirely unite these data with the irrigation data, but we believe that this is a very interesting observation that warrants reporting. For example, what if irrigation is a sub-optimal mode of scab suppression due to a reduction in *Pseudomonas* biocontrol strains? Our data potentially supports improved suppression strategies where irrigation is combined with the application of biocontrol strains to maximise the effect of a protective microbiome.

In terms of manuscript length, we agree to an extent with the reviewer’s comments, especially as we have had to lengthen some sections to include new results, methods and citations. We have therefore made edits throughout the manuscript (and figures) to reduce their length, and have removed most of the section that relates to the suppression of other plant pathogens (the old Figure 7 plus associated main text, as recommended by this reviewer). These edits are summarised in a “track changes” version of the manuscript.

We believe that these edits help the manuscript flow more effectively. However, we are keen that the manuscript properly reflects the diverse experimental approaches that were undertaken in this study, which does require the manuscript to be relatively large. We also strongly believe that extensive supplementary information is highly beneficial for other researchers and fits with the ethos of *eLife*. We want to be as open as possible in presenting associated data, such as the visualisation of gene clusters, carefully annotated mass spectrometry and NMR data, detailed figures showing phenotypic results, and extra data that supports key findings in the main manuscript text.

Unless I missed the point (and I would gladly read an appeal by the authors or other reviewers if I did), the storyline doesn't lead to a logical conclusion. The authors show multiple experiments of notable quality, but I find it hard to describe exactly what the authors found in relation to their stated problem. Were they able to provide insight into the irrigated vs. unirrigated phenomena? Was the in-vitro phenotyping relevant to the field?

Our new experiments provide substantial insight to this question. We now have in planta data that relates the in vitro phenotyping to real biocontrol properties. The relevance to the unexpected irrigation data is discussed above. It is also worth emphasising the numerous take-home messages that arise from this single study:

a) Correlation of phenotype and genotype across a highly diverse pan-genome leads to mechanistic understanding of plant pathogen suppression. This is validated by genetics, chemical analysis, in vitro experiments and new in planta experiments.

b) There is huge genetic and phenotypic diversity within a single bacterial genus in a single field, which is highly dynamic and is hidden by conventional amplicon sequencing approaches. The data we report is therefore useful for researchers ranging from natural product discovery to soil ecology.

c) We provide the first demonstration that an isolated *Pseudomonas* cyclic lipopeptide is active towards *S. scabies*.

d) Despite their reputation as biocontrol strains producing multiple bioactive molecules, a large proportion of *Pseudomonas* strains are non-suppressive in vitro. Our study should prompt some interesting follow-on questions about the role and relative dominance of these non-suppressive strains.

Unfortunately, while I find the paper interesting, it isn't currently clear to me that the authors made a discovery that would merit publication in eLife. I urge the authors to try and distill the discoveries made in this study into a more concise and logically appealing report.1. The authors took a genomic approach alongside an array of classical microbiological techniques to study the potential interactions between Pseudomonads and Streptomyces scabie under defined laboratory conditions. They reached the conclusion that CLPs and HCN can help Pseudomonads to inhibit and outcompete S. scabie. However, while these results provide well-demonstrated phenotypes, considering the large body of work on Pseudomonad-dependent biocontrol it is not surprising that toxic metabolites like HCN can inhibit competitor growth. While CLPs provide a novel result, the authors do not isolate the effect of CLPs as compared with motility effects – for example, they do not show that the purified CLP can cause these effects.

We have now purified a CLP from strain Ps682. We had previously shown that production of this viscosin-like CLP was critical for motility and biological activity. This CLP (an isomer of viscosin: “viscosin I”) has been characterised in detail by NMR and further MS/MS experiments. Viscosin I inhibits *S. scabies* in disk diffusion assays, which has been incorporated into Figure 5.

2. The authors choose to focus on Pseudomonads, which have been already extensively implicated in biocontrol in the last 3 decades, with many studies linking various secondary metabolites to disease control in various crops, mainly in the context of fungal infections, as well as in the context of aridity and irrigation: the authors are should be aware of the multiple papers from the Thomashow's lab on irrigations and biocontrol by Pseudomonads, which are not citated here… (Mavrodi 2012 and 2018 for example).

Apologies – we were aware of this work and the absence of citations was a mistake. These are now cited along with a sentence of descriptive text (lines 515-517).

3. Since all of the phenotypic work in the study is done on agar plates and not on lab grown plants or in the field its completely feasible that the inferred interactions do not take place outside of the lab. This is a major limitation of the current study and a reasonable explanation to their later contradictory findings.

This has been discussed in detail above.

4. Ultimately, using both metagenomics and culturing the authors find no substantial difference in the microbial communities of irrigated vs. unirrigated fields. Thus, the protective effects of irrigation remain a mystery and Pseudomonads have again been shown to inhibit pathogens in-vitro. There could be many possible reasons for this discrepancy. Possibly another suppressive type of bacteria or fungi or maybe just the abiotic conditions imposed under irrigation (lower oxygen levels for example) shape the behavior of the pathogen itself.

This has been discussed above. We have also included a statement in the Discussion on the possibility that irrigation reduces the relative fitness of *S. scabies* versus *Pseudomonas* spp. yet may not favour an optimal microbiome for disease suppression based on our in planta data. The data we report also provides an ideal foundation for ourselves and others to investigate this observation in more detail (for example, how and why does the environment shape the *Pseudomonas* population? Is this observed for other diverse genera?). This manuscript represents one of the most extensive attempts to map intra-genus variation in such defined conditions. Comparable studies linking microbial genotype and biological activity either do not contain an explicit link to environmental sampling (e.g. *Nature Microbiology*, 2019, 4, 996) or focus on a specific environment but do not investigate the link between biological activity and plant health (e.g. *Nature Microbiology*, 2018, 3, 909).

Reviewer #2:The manuscript by Stefanato et al. constitutes a brilliant multidisciplinary study of the pan-genomic variation in molecular interactions with plant pathogens, which sheds light on their potential plant-protective role within plant microbiomes. The research includes large-scale genome sequencing and analysis, mass spectrometry, in vitro bioassays and ecological analysis. The discovery that cyclic lipopeptides inhibit bacterial plant pathogens is very novel and exciting, and the general approach provides a blueprint for future pan-genomic analyses of host-microbe interactions, also outside the plant microbiome field. In general, I am of the opinion that this work is very suitable for publication in eLife.However, I do have a number of concerns that I would like to see addressed:– The quality of the phylogenetic tree used as reference for the analysis is quite important for the rest of the analysis. Given that single marker genes were used, is there a specific reason why the authors chose to use FastTree instead of a maximum likelihood algorithm like IQTree or RAxML(-NG)? Unlike the authors mention in the methods, FastTree itself is not a maximum likelihood algorithm but an 'approximate maximum likelihood' algorithm. Also, it would be very useful if bootstrap information would be provided, at least somewhere in the supplement or in an online data file.

The tree has now been generated using RAxML and bootstrap values are reported in a tree in the Supplementary Information (Figure S4). In terms of the associated informatic data, we have also repeated the associated antiSMASH 5 analysis (conducted Jan 2021) to ensure that the biosynthetic gene cluster information is as accurate as possible.

– The authors note that they find some BGCs that are 'rarely found in pseudomonads'. Given that draft genomes were used, did the authors make sure that there was no contamination of the genome assemblies by co-isolated strains? A scary amount of draft genomes in the public databases contain contigs from co-isolates or contaminant strains. A simple CheckM analysis (https://genome.cshlp.org/content/25/7/1043.full) would suffice to rule this out.

Thank you for the suggestion of CheckM. We have now checked each genome and there is no evidence of contamination. CheckM analysis of duplicated housekeeping genes instead identified assembly errors in two strains, where contigs were duplicated in their sequence files (for example, two different contigs were effectively identical in strain Ps861). This has now been corrected in the online sequences. We should note that we did originally manually review gene cluster data for evidence of BGCs coming from non-*Pseudomonas* strains by assessing features such as GC content and sequence identity to know BGCs. More generally, there is strong evidence of horizontal transfer of BGCs between diverse bacterial taxa, including pseudomonads (e.g. *Appl. Environ. Microbiol.*, 2018, 84, e02828–17; *Appl. Environ. Microbiol.*, 2017, 83, e01169–17).

– With regard to the correlation analysis, it should be noted that this can be highly biased by the phylogenetic structure of the data, as well as isolation bias (having isolated more closely related strains from a taxon leads to higher correlations to elements specific to that taxon). While this is an inherent limitation, it should be kept in mind. Specifically, I believe that the Chi-squared test (mentioned in Figure 3) is, strictly speaking, not appropriate, as it assumes independence of the observations, which is not the case for gene clusters found in strains that are related to each other through recent common ancestry. Regardless, the calculations are, in practice, still able to identify the patterns that are also visible simply by eye in Figure 2, but the statistics may not be so relevant.

We appreciate the feedback over potential biases introduced by the phylogenetic structure of the data. This was one reason why we functionally tested some of the most prominent correlations via gene deletions and tried to highlight this in the text of the manuscript. It may be difficult to separate phylogenetic structure from BGC presence/absence given that the acquisition of a specific BGC may be a critical determinant of lifestyle and thus shape the evolution of specific clades. For example, CLPs aid motility and suppress a variety of organisms. In relation to the Chi-squared tests in Figures 3 and 8, we have since reasoned that a Chi-squared test was not the most suitable statistical test, as it assesses numerical data instead of purely categorical data (i.e., a score of 3 is more similar to 2 than to 0, which is not reflected by a Chi-squared test that assesses for the distribution of categorical variables). We have revised this figure to show statistical significances from a MannWhitney test, which is used to assess for numerical differences between two groups.

– For the new viscosin and tensin analogues, it is not clearly indicated in the text/figures how they are related to known viscosins/tensins. 'Viscosin-like' seems to imply that it concerns a new variant, but the supplementary figure suggests otherwise.

We have endeavoured to make this clearer in the text and figures. For tensin and other molecules identified by mass spectrometry, the key message was that our MS data and BGC analysis was fully consistent with tensin (and other CLPs in other strains), but without an available authentic standard, we could not explicitly confirm this. For example, simply one stereocentre could be different, such as L-alloisoleucine instead of L-isoleucine, but we believe this is beyond the scope of this paper. For viscosin, the availability of *P. fluorescens* SBW25 enabled a direct analytical comparison, which indicated that while all BGCs were very similar to each other (>90% identity) and MS data was almost identical, the different LC retention time of some viscosin-like CLPs (e.g. from Ps682) meant that they are isomers of the characterised viscosin. We have now isolated and characterised the viscosinlike CLP from Ps682 using NMR and MS, which highlights possible structural differences but does demonstrate that its chemical connectivity is identical to viscosin and a related molecule called WLIP.

– The authors conclude that a subset of the pseudomonads function as generalist pathogen suppressors, based on in vitro experiments, and also imply that pseudomonads use CLPs and HCN to inhibit pathogenic streptomycetes. However, it should be noted that in vitro activities may not be representative of in vivo ecological functions, as, in planta, the required molecular triggers to induce the expression of the relevant gene clusters may not be present (in sufficient amounts). I.e., it might well be that in planta, the BGCs are only actively inhibiting a smaller subset of the pathogens than the ones against which their products are active in vitro. I believe this caveat should be clearly indicated in the Discussion section.

We think the new in planta data addresses this comment. This shows that the Ps619 CLP is a key determinant of biocontrol in plant experiments. We actually observed examples of the trait suggested by the reviewer, where HCN did not have a clear role for Ps619 biocontrol, while the in vitro suppressor Ps682 was not active in planta. As suggested, the caveat of whether a molecule is produced (or not) during plant colonisation is discussed in the manuscript.

Reviewer #3:Overall, this is a very interesting study where integrated genomics, metabolomics, phenotypic analysis and molecular biology approaches were used to identify the genetic determinants of Pseudomonas antagonism towards Stretomyces scabies causing the common scab disease of potato. A comparative approach was used where Pseudomonas associated with an irrigated field (where less symptoms of common scab are generally observed) and a non-irrigated field (where more disease pressure is observed) were compared. The amount of work invested here is important, the analyses are sound and the paper is overall well written.Main concerns:1) The paper (including the title) should more focused on common scab, while the other two pathogens used in parallel should be more accessory (validation tools only).

We have moved the original Figure 7 (related to inhibition of other pathogens) to the SI and have reduced the associated text substantially.

2) The introduction should be significantly improved (and more accurate) to better reflect the recent literature on Pseudomonas against S. scabies and also on the genomics of phytobeneficial Pseudomonas. For example, Pseudomonas have been successfully used under soil (Arseneault A, C Goyer, M Filion. 2016. Biocontrol of potato common scab is associated with high Pseudomonas fluorescens LBUM223 populations and phenazine-1-carboxylic acid biosynthetic transcripts accumulation in the potato geocaulosphere. Phytopathology 106: 963-970.) and field conditions to control common scab (Arseneault A, C Goyer, M Filion. 2015. Pseudomonas fluorescens LBUM223 increases potato yield and reduces common scab symptoms in the field. Phytopathology 105: 1311-1317.). Also, I invite the authors to consult a recent paper on the genomics of phytobeneficial Pseudomomas (Biessy A, A Novinscak, J Blom, G Léger, LS Thomashow, FM Cazorla, D Josic, M Filion. 2019. Diversity of phytobeneficial traits revealed by whole-genome analysis of worldwide-isolated phenazine-producing Pseudomonas spp. Environmental Microbiology. 21: 437-455.), which provides valuable information that will guide the authors to modify their introduction section. The authors seem to be unaware of a significant portion of the literature important for their study.

We definitely should have included these references and were aware of them – they are all relevant and are now cited. We had originally included Arseneault 2013 as an example of *Pseudomonas* sp. LBUM223 biological activity, but agree that the follow-up papers are important citations.

3) It is not clear to me as to how the screening of all the isolated Pseudomonas was performed to avoid redundancy. The authors claim that 240 strains of Pseudomonas were isolated. Does this mean that these 240 strains are different or that some of these strains are in fact identical. Less than a third of these strains were sequenced, what about the others?

We cannot be certain that every one of those 240 strains was unique. In fact, the detection of substantial clonal populations would have told a different story, which would have been interesting in itself (for example, that the same field and/or plants selects for a very narrow population of pseudomonads). However, our initial phenotypic scoring (shown as a sheet in Supporting Dataset 1) provided evidence that the strains were phenotypically diverse, while genome sequencing showed that no two strains are identical, although some are clearly closely related. Apart from the criterion of selecting an approximately equal number of suppressive and non-suppressive strains across multiple soil and plant samples (from the same field), bacteria were selected randomly for genome sequencing, with the caveat that strains from the same sample site with the same phenotype score pattern were avoided wherever possible.

4) As the main conclusion seems to be that the Pseudomonas capable of producing CLP and hydrogene cyanide are the most antagnostic against S. scabies, why not perform an in planta experiment showing that at least one wildtype hydrogence cyanide and CLP producer can reduce common scab symptoms (while isogenic mutants for these genes are less efficient). It is surprising to see that these organisms are less present in the irrigated field where less disease is observed. An in planta confirmation would significantly strengthen the conclusion of this study.

This has been discussed in detail above. We believe that this reviewer concern has been directly addressed by potato scab biocontrol experiments that demonstrate the effect described by the reviewer.

[Editors’ note: what follows is the authors’ response to the second round of review.]

The reviewers have discussed their reviews with one another, and the Reviewing Editor has drafted this to help you prepare a revised submission.The extensive additional work and in planta experiments have addressed previous concerns and greatly improved the work. While the reviewers are enthusiastic about the current version, the authors should still address the following suggestions, raised by the third reviewer, in order to improve the clarity of the manuscript prior to publication.Essential revisions:Suggestions to improve clarity and readability.1. There is growing body of literature showing that drought enriches for commensal Streptomyces (and dramatically depletes Proteobacteria) in the rhizosphere, which is reversible after irrigation (i.e. https://doi.org/10.1073/pnas.1717308115; https://doi.org/10.1038/s41477-021-00967-1) (2021). Although these studies are from the perspective of changes that are protective against drought, it seems the same mechanism that results in enrichment of Streptomyces (and depletion of Proteobacteria) during drought may deplete Streptomyces (and enrich Pseudomonas) during irrigation. It would be useful for the authors to compare the changes they see after irrigation to thoseinduced by drought in the literature.

While we do see a modest positive change in *Pseudomonas* abundance on irrigation, in agreement with the papers cited here, we did not detect a major difference in *Streptomyces* abundance between our irrigated and unirrigated field samples. This is perhaps not surprising as our experimental setup was not aiming to simulate drought, and the unirrigated samples still received enough water for plants to grow healthily. We have incorporated the references above into the Discussion section, where we present an alternative hypothesis for our data based on irrigation induced changes in overall microbial abundance, as requested for point 2.

2. I am not sure that the data that led to the conclusions that "irrigation led to a decrease in the proportion of suppressive pseudomonads on potato roots" (568-9) is really rigorous enough to support this counter intuitive conclusion. The sample sizes of suppressive isolates is quite small, and it is hard to be certain that culturing really represents a random and representative sample of what is in the soil. In retrospect, it seems like it would have been more straightforward to perform qPCR to look at abundance of genes or transcripts related to HCN or CLPs in the two soil types. However, as the authors found an increase in the total abundance of Pseudomonas after irrigation (Figure 1C), I think they could make a stronger case based on my point above and speculate that an increase in total Pseudomonas might contribute to suppression, rather than a change in function of Pseudomonas (which is not really supported by the data).

We respectfully disagree that our root sampling is not representative of what is going on in the soil. The results of two independent field experiments conducted two years apart both show a decrease in the proportion of suppressive pseudomonads associated with potato roots. The first experiment (Figures 1-2) contained too few sequenced isolates to draw robust conclusions, therefore we designed the second experiment (Figure 8) to investigate this phenomenon further. This experiment showed highly significant differences in the proportion of suppressive isolates on irrigated and non-irrigated roots. While there will inevitably be a degree of sampling bias in any experiment based on culturable isolates, we did our best to ensure that any bias would apply equally to all samples (soil/roots and irrigated/non-irrigated). Our previous expt. showed that a broad distribution of different *Pseudomonas* genotypes could be cultured from the soil by the methods used here (Figure 2), so we are confident that what we show in Figure 8 is a real phenomenon.

That said, the hypothesis presented by the reviewers; that relative *Pseudomonas* and *Streptomyces* abundance are affected by irrigation and this may be the primary factor in scab suppression, is a valid one. We have amended the discussion to include this as an alternative explanation for our results, and include the suggested references in support of this.

3. For Figure 2, it would be helpful if the authors could label each Pseudomonas subgroup as in Garrido-Sanz et al. and/or include some widely studied or type strains to help orient the reader and provide context for the strains and tree. This would also help others in the field use this as a resource to identify functional genes that are Pseudomonas-clade specific.

Thank you for the suggestion. We have taken a slightly different strategy as we were wary of adding too much information to a figure that was already very complex. We agree that the relationship to reference strains is useful information, and we had already presented the *Pseudomonas* sub-groups and reference strains in Figure 2 —figure supplement 3. This figure is also now referred to in the Figure 2 legend. In addition, a link to an interactive online tree has also been added (via iTOL, https://itol.embl.de/tree/902431818658671633339579), which features reference strains and all phenotypic and genotypic information presented in Figure 2. We hope this online tree serves as a resource for other researchers.

4. For the paragraphs starting on line 192 (mostly related to Figure 2) please ensure you reference the figure or data in each instance. For instance, "Multiple BGCs were commonly found across the sequenced strains, including BGCs predicted to make CLPs (42), arylpolyenes (43) and HCN (38)." This is shown in Figure 2, but it is not referenced in the text so it was hard for me find.

Figure 2 and Supplementary File 1 are now referenced towards the beginning of this paragraph, which should guide the reader wanting to see the strains that have these BGCs.

5. There are multiple auxin biosynthesis pathways in Pseudomonas (IoaX/IAN, IAH, etc.; for instance, see https://journals.plos.org/plospathogens/article?id=10.1371/journal.ppat.1006811) so it would be helpful to know if the genes identified are all the same pathway. I'm asking in part because it's a bit surprising that IAA biosynthesis is so uncommon in your analysis while other studies have found it to be quite prevalent (for instance, while this study is limited to P. brassicearum, they identified IAA biosynthesis in the whole clade https://www.microbiologyresearch.org/content/journal/jmm/10.1099/jmm.0.001145)

The methodology we used for identifying all genes and gene clusters is reported in detail in Appendix 1. To identify IAA BGCs, we searched for clustered homologues of IaaM and IaaH from *Pseudomonas* savastanoi. We therefore appreciate the reference to another well-characterised route to auxin in a *Pseudomonas* species. Using the information in the PLOS Pathogens paper mentioned above, we assessed our strains for AldA, which was reported to be the critical indole-3-acetaldehyde dehydrogenase for IAA biosynthesis in *Pseudomonas* syringae DC3000. This revealed that homologues of this protein are encoded in all of the strains reported in our study (>88% identity) and are found in exactly the same genetic context as in DC3000. This provides confidence that these homologous proteins represent a genuine route to IAA. We distinguish this route from the other IAA route by naming them IAA 1 and IAA 2 (equivalent to the nomenclature we use for PQQ BGCs). All relevant information has been updated (additional citation, Figure 2 details, Supplementary File 1, Appendix 1 details on AldA), and the manuscript text has been adjusted accordingly.